# CoCoA-Mix: Confusion-and-Confidence-Aware Mixture Model for Context Optimization

**Dasol Hong** [1]   **Wooju Lee** [1]   **Hyun Myung** [1]

## Abstract

Prompt tuning, which adapts vision-language models by freezing model parameters and optimizing only the prompt, has proven effective for task-specific adaptations. The core challenge in prompt tuning is improving specialization for a specific task and generalization for unseen domains. However, frozen encoders often produce misaligned features, leading to confusion between classes and limiting specialization. To overcome this issue, we propose a confusion-aware loss (CoA-loss) that improves specialization by refining the decision boundaries between confusing classes. Additionally, we mathematically demonstrate that a mixture model can enhance generalization without compromising specialization. This is achieved using confidence-aware weights (CoA-weights), which adjust the weights of each prediction in the mixture model based on its confidence within the class domains. Extensive experiments show that CoCoA-Mix, a mixture model with CoA-loss and CoA-weights, outperforms state-of-the-art methods by enhancing specialization and generalization. Our code is publicly available at https://github.com/url-kaist/CoCoA-Mix.

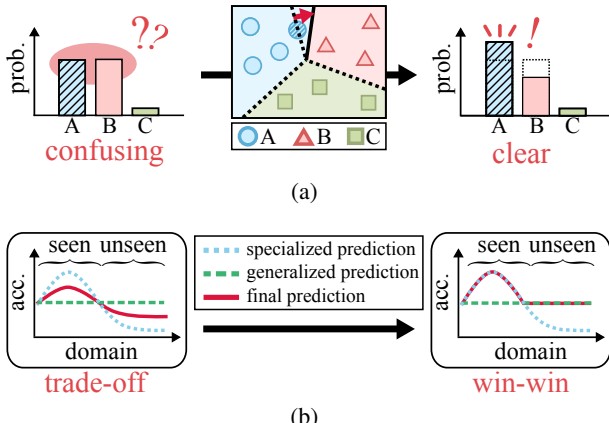

(a)

(b)

Figure 1: (a) Effect of CoA-loss. The left and right sides show probabilities for the hatched sample on the decision boundary. In the middle figure, dashed and solid lines indicate the current and updated decision boundaries. CoA-loss assigns higher weights to this confusing sample, enhancing specialization in prompt tuning. (b) Effect of CoA-weights. The left and right sides show performance across domains. CoA-weights increase confidence in the specialized prediction for seen domains while reducing it for unseen domains, preserving both specialization and generalization.

## 1. Introduction

Pre-trained vision-language models (VLMs) have achieved remarkable results in diverse downstream tasks, such as image classification (Fu et al., 2022), object detection (Zhong et al., 2022; Gu et al., 2021), and visual question answering (Cho et al., 2021; Lin et al., 2022). These models align visual and textual embeddings through extensive pre-training

on large-scale datasets, enabling remarkable zero-shot capabilities. However, their reliance on generic embeddings often limits their effectiveness in task-specific applications.

To mitigate this limitation, prompt engineering has emerged as a practical way to enhance performance on a specific task (Wang et al., 2021; 2022a). The method involves manually designing a task-specific text template for the model input. For example, in image classification, hand-crafted prompts such as `"a photo of a [CLASS], a type of flower"` help associate visual embeddings with the correct class labels, such as "hibiscus" or "sword lily." (Zhou et al., 2022c) Although effective, the manual process is labor-intensive and requires domain expertise, limiting scalability across diverse tasks.

Prompt tuning has become a scalable alternative to manual prompt engineering by replacing hand-crafted prompts with

---

[1]Urban Robotics Lab, School of Electrical Engineering, Korea Advanced Institute of Science and Technology, Republic of Korea. Correspondence to: Dasol Hong <ds.hong@kaist.ac.kr>, Wooju Lee <dnwn24@kaist.ac.kr>, Hyun Myung <hmyung@kaist.ac.kr>.

*Proceedings of the 42nd International Conference on Machine Learning*, Vancouver, Canada. PMLR 267, 2025. Copyright 2025 by the author(s).

learnable prompts (Zhou et al., 2022c; Zhu et al., 2023; Zhou et al., 2024; Zhang et al., 2024a). This method allows VLMs to effectively align textual embeddings with the corresponding visual embeddings by optimizing the prompts for a specific task while freezing the model parameters.

However, the existing methods have two limitations. First, existing methods do not explicitly address confusing cases arising from the frozen visual encoder. The frozen visual encoder often fails to capture task-specific features, struggling to distinguish between different classes. However, most existing methods rely on standard cross-entropy loss, which is ineffective in handling confusing cases and consequently limits their specialization (Zhou et al., 2022c;b). Second, achieving generalization to unseen domains is crucial in prompt tuning, but current methods often sacrifice specialization to improve generalization. Most studies inherently assume specialization and generalization are competing objectives. As a result, the trade-off problem remains an open challenge.

To overcome these limitations, we propose a confusion-and-confidence-aware mixture model (CoCoA-Mix), which combines confusion-aware loss (CoA-loss) and confidence-aware weights (CoA-weights). We first introduce a mixture model that combines predictions from individual prompts, providing a theoretical framework to analyze prompt tuning in terms of specialization and generalization. Building on this theoretical insight, CoA-loss improves specialization by applying larger gradient to confusing cases, refining decision boundaries as shown in Figure 1(a). Then, CoA-weights achieve generalization without compromising specialization by scaling mixture model weights based on the confidence of individual prompts across class domains, as shown in Figure 1(b). As a result, our method enhances both specialization and generalization by ensuring that the error of the mixture model remains lower than the minimum error of individual prompts. Main contributions are as follows:

- We provide a mathematical framework demonstrating that specialization and generalization can be improved simultaneously.

- We propose a CoCoA-Mix framework, consisting of CoA-loss and CoA-weights. CoA-loss boosts specialization by improving classification for confusing cases, while CoA-weights improve generalization by adjusting the weights of individual prompts in the mixture model based on their confidence over class domains.

- The proposed method achieves average harmonic mean improvements of $15.28\%$ and $3.28\%$ over zero-shot CLIP in base-to-new generalization and cross-dataset transfer, respectively; it also improves the average accuracy in few-shot class-incremental learning by $5.6\%p$.

## 2. Related Work

### 2.1. Textual Prompt Tuning

Prompt tuning has emerged as a method to reduce the reliance on human expertise while improving the performance of VLMs on specific tasks. The method can be categorized into visual prompt tuning (Jia et al., 2022; Bahng et al., 2022), textual prompt tuning (Zhou et al., 2022c;b; Zhu et al., 2023; Yao et al., 2023; Zhang et al., 2024b), and visual-textual prompt tuning (Khattak et al., 2023a;b). In this paper, we focus on textual prompt tuning, which replaces hand-crafted prompts with learnable prompts and optimizes them using few-shot training data, while keeping the model parameters frozen.

### 2.2. Loss for Prompt Tuning

In textual prompt tuning, the frozen encoders map inputs to generic embeddings, which can lead to misaligned vision-text embeddings for specific tasks. A common strategy to mitigate this issue is to employ a standard cross-entropy loss that aligns textual prompts with their corresponding visual embeddings. CoOp (Zhou et al., 2022c) pioneered textual prompt tuning by optimizing prompts for specific tasks using cross-entropy loss. Subsequent works employed regularization via hand-crafted prompts, which enhanced specialization by preventing prompts from learning unintended patterns (Zhu et al., 2023; Yao et al., 2023; Zhang et al., 2024b). However, such regularization methods may prevent prompts from fully capturing task-specific patterns, limiting specialization in complex tasks. Other approaches (Khattak et al., 2023a; Zhang et al., 2024b) introduce additional network parameters to enhance the representational capacity of the model. MaPLe (Khattak et al., 2023a) leverages visual-language interactions via a coupling function, while DePT (Zhang et al., 2024b) employs a dual-head architecture to decouple task-specific and task-shared knowledge into separate feature spaces. Although these methods improve generalization, they increase the number of learnable parameters, leading to overfitting when training data is scarce and limiting scalability. In contrast, we aim to enhance both specialization and generalization without introducing additional network components.

### 2.3. Mixture of Prompts

Prompt ensembling has been studied to improve generalization in VLMs. Allingham et al. (2023) propose zero-shot prompt ensembling (ZPE), which automatically assigns weights to hand-crafted prompts from a large pool, leading to improved zero-shot accuracy. Lu et al. (2024) extend this to model-level ensembles suited to varying resource settings, but this requires multiple forward passes at inference time. Despite their generalization benefits, neither

approach explicitly addresses the challenge of achieving task-specific specialization. In contrast, our method jointly improves specialization and generalization, while remaining computationally efficient by avoiding multiple forward passes.

# 3. Proposed Method

## 3.1. Preliminary

In downstream tasks, CLIP employs visual and textual encoders to map images and textual prompts into a shared embedding space. The textual prompts $t$ can be generated as hand-crafted prompts by embedding class labels into predefined templates, such as `"a photo of a [CLASS]."` With $t$, the probability of the image belonging to the class $l$, $\hat{p}_t(l)$, is defined as follows:

$$\hat{p}_t(l) = \frac{\exp\left(s_t(l)/\tau\right)}{\sum_{l' \in \mathcal{Y}} \exp\left(s_t(l')/\tau\right)}, \tag{1}$$

where $\tau$ is the temperature scale, $\mathcal{Y}$ represents the set of classes, and $s_t(l)$ denotes the cosine similarity between the visual and textual embeddings of the class $l$ generated by $t$.

## 3.2. Decomposing Specialization and Generalization

The expected error $\epsilon_T(\hat{p})$ of a predictive distribution $\hat{p}$ in an arbitrary target domain $\mathcal{D}_T$ is defined using the Kullback-Leibler (KL) divergence as follows:

$$\epsilon_T(\hat{p}) = \mathbb{E}_{(\mathbf{x},y)\sim\mathcal{D}_T}\left[-\log\hat{p}(y)\right], \tag{2}$$

where $y$ is the ground-truth label for the image $\mathbf{x}$.

We introduce a mixture model to derive an upper bound for the error in the target domain, providing a framework for analyzing specialization and generalization.

**Definition 3.1** (Mixture model). Let $K+1$ different prompts be given by $\mathcal{T} = \{t_0, t_1, \cdots, t_K\}$, and let $\boldsymbol{\pi} = \{\pi_0, \pi_1, \cdots, \pi_K\}$ denote a set of non-negative weights satisfying $\sum_{i=0}^{K} \pi_i = 1$. The mixture model $\hat{p}_{\mathcal{T}}^{\boldsymbol{\pi}}$ is defined as a weighted combination of the individual prompts:

$$\hat{p}_{\mathcal{T}}^{\boldsymbol{\pi}}(l) = \frac{\exp\left(\sum_{i=0}^{K} \pi_i s_{t_i}(l)/\tau\right)}{\sum_{l' \in \mathcal{Y}} \exp\left(\sum_{i=0}^{K} \pi_i s_{t_i}(l')/\tau\right)}. \tag{3}$$

**Theorem 3.2.** *The expected error of the mixture model $\hat{p}_{\mathcal{T}}^{\boldsymbol{\pi}}$ can be bounded as follows:*

$$\epsilon_T(\hat{p}_{\mathcal{T}}^{\boldsymbol{\pi}}) \leq \sum_{i=0}^{K} \pi_i \epsilon_T(\hat{p}_{t_i}). \tag{4}$$

*The proof is provided in Section A.*

**Lemma 3.3.** *Let the class set of the target domain $\mathcal{D}_T$ be partitioned into $K+1$ disjoint subsets, with corresponding sub-domains $\mathcal{D}_{T_0}, \mathcal{D}_{T_1}, \cdots, \mathcal{D}_{T_K}$, such that $\mathcal{D}_T = \bigsqcup_{i=0}^{K} \mathcal{D}_{T_i}$. Then, the expected error of $\hat{p}_{\mathcal{T}}^{\boldsymbol{\pi}}$ is given by:*

$$\epsilon_T(\hat{p}_{\mathcal{T}}^{\boldsymbol{\pi}}) = \sum_{i=0}^{K} \lambda_i \epsilon_{T_i}(\hat{p}_{\mathcal{T}}^{\boldsymbol{\pi}}), \tag{5}$$

*where $\lambda_i = \mathrm{Pr}_{(\mathbf{x},y)\sim\mathcal{D}_T}\left[(\mathbf{x},y) \in \mathcal{D}_{T_i}\right]$ denotes the probability that a sample from the target domain $\mathcal{D}_T$ belongs to the sub-domain $\mathcal{D}_{T_i}$, satisfying $\sum_{i=0}^{K} \lambda_i = 1$. Based on Theorem 3.2, the error of the mixture model in the arbitrary target domain can be upper-bounded as follows:*

$$\epsilon_T(\hat{p}_{\mathcal{T}}^{\boldsymbol{\pi}}) \leq \sum_{i=0}^{K} \lambda_i \left( \pi_i^{in} \underbrace{\epsilon_{T_i}(\hat{p}_{t_i})}_{\substack{specialization \\ error}} + \underbrace{\sum_{\substack{j=0 \\ j\neq i}}^{K} \pi_j^{out} \epsilon_{T_i}(\hat{p}_{t_j})}_{\substack{generalization \\ error}} \right), \tag{6}$$

*where $\pi_i^{in}$ denotes the mixing weight of the prompt $t_i$ for its own domain $\mathcal{D}_{T_i}$, and $\pi_j^{out}$ denotes the mixing weight of the prompt $t_j(j \neq i)$ when applied to the domain $\mathcal{D}_i$. Here, $\pi_i^{in} + \sum_{\substack{j=0 \\ j\neq i}}^{K} \pi_j^{out} = 1$.*

Let $\mathcal{Y}$ denote the set of all classes in the arbitrary target domain $\mathcal{D}_T$, which is partitioned as $\mathcal{Y} = \bigsqcup_{i=0}^{K} \mathcal{Y}_i$. Specifically, we assume that labeled data is provided for $\mathcal{Y}_1, \cdots, \mathcal{Y}_K$, whereas no supervision is available for $\mathcal{Y}_0$. Due to the absence of labeled data in $\mathcal{D}_{T_0}$, the associated prompt $t_0$ cannot be specialized through training. Instead, we utilize a generalized hand-crafted prompt such as `"a photo of a [CLASS]."`

Building on this, we detail the proposed method for specialization and generalization in prompt tuning. In Section 3.3, we propose confusion-aware loss (CoA-loss), which focuses on confusing cases in prompt tuning and effectively reduces the *specialization* error $\epsilon_{T_i}(\hat{p}_{t_i})$ for each sub-domain $\mathcal{D}_{T_i}$. Subsequently, Section 3.4 enhances *generalization* by optimizing confidence-aware weights (CoA-weights) based on the confidence of each prompt on the given class domain, tightening the upper bound of $\epsilon_T(\hat{p}_{\mathcal{T}}^{\boldsymbol{\pi}})$. As a result, the proposed method enhances specialization and generalization simultaneously. Figure 2 shows the overall framework of the confusion-and-confidence-aware mixture model (CoCoA-Mix), which combines CoA-loss and CoA-weights.

## 3.3. Confusion-Aware Loss for Specialization

Let $\hat{p}$ be a predictive distribution and $\mathcal{D}_S$ be the source domain. According to Nguyen et al. (2022), the expected

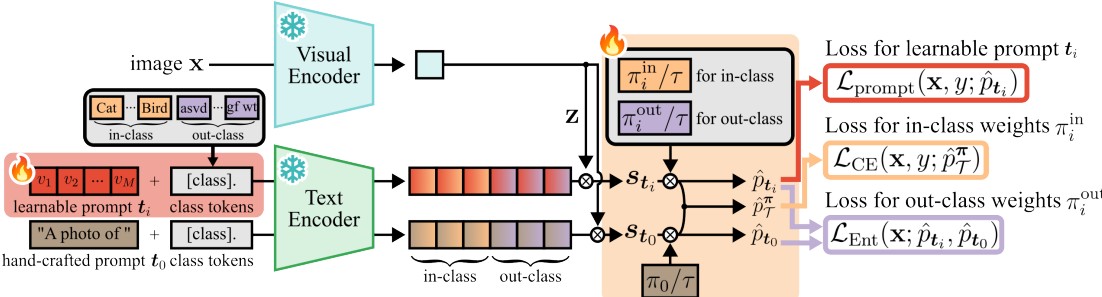

Figure 2: CoCoA-Mix framework integrates confusion-aware loss (CoA-loss) for specialization and confidence-aware weights (CoA-weights) for generalization, ensuring performance improvements without trade-offs. The learnable prompt $t_i$ is optimized with CoA-loss to specialize in distinguishing confusing classes within the training domain. CoA-weights adjusts prediction confidence by increasing $\pi_i^{\text{in}}$ for in-class and decreasing $\pi_i^{\text{out}}$ for out-class. At inference, the specialized predictions $\hat{p}_{t_i}$, adjusted via CoA-weights, are combined with the generalized predictions $\hat{p}_{t_0}$ to ensure generalization while preserving specialization.

error $\epsilon_T(\hat{p})$ on the target domain $\mathcal{D}_T$ is bounded as follows:

$$\epsilon_T(\hat{p}) \leq \epsilon_S(\hat{p}) + \frac{C}{\sqrt{2}} \sqrt{\text{KL}(p_T(\mathbf{z})|p_S(\mathbf{z})) + \delta}, \quad (7)$$

where $\mathbf{z}$ is defined as a visual embedding; $C$ is a constant that bounds $\log \hat{p}(l)$, ensuring each class probability is at least $\exp(-C)$; $p_T$ and $p_S$ are the marginal distribution of $\mathbf{z}$ for the target and source domains, respectively; and $\delta$ denotes the conditional misalignment $\mathbb{E}_{p_T(\mathbf{x})}[\text{KL}(p_T(y|\mathbf{x})|p_S(y|\mathbf{x}))]$, which is typically small.

We focus on textual prompt tuning, which cannot optimize the visual embedding $\mathbf{z}$ due to the frozen encoders. Thus, Equation (7) suggests that minimizing $\epsilon_S(\hat{p})$ can reduce $\epsilon_T(\hat{p})$, enabling error minimization in the mixture model via source domain *specialization*.

Most existing methods use standard cross-entropy, defined as follows, for the specialization in prompt tuning:

$$\mathcal{L}_{\text{CE}}(\mathbf{x}, y; \hat{p}_t) = -\log \hat{p}_t(y). \quad (8)$$

Cross-entropy assigns gradients based on class probabilities, assigning larger gradients to misclassified samples. While effective for misclassified cases, it relies only on individual probabilities and does not consider inter-class relationships, limiting its ability to handle class confusion. Addressing confusing classes is crucial in prompt tuning with limited training data because they significantly impact the decision boundary. To overcome this limitation, we propose confusion-aware loss (CoA-loss), defined as follows:

$$\mathcal{L}_{\text{CoA}}(\mathbf{x}, y; \hat{p}_t) = 1 - \hat{p}_t(y). \quad (9)$$

The overall loss $\mathcal{L}_{\text{prompt}}$ for optimizing the prompt $t$ is given by:

$$\mathcal{L}_{\text{prompt}}(\mathbf{x}, y; \hat{p}_t) = \mathcal{L}_{\text{CE}} + w\mathcal{L}_{\text{CoA}}, \quad (10)$$

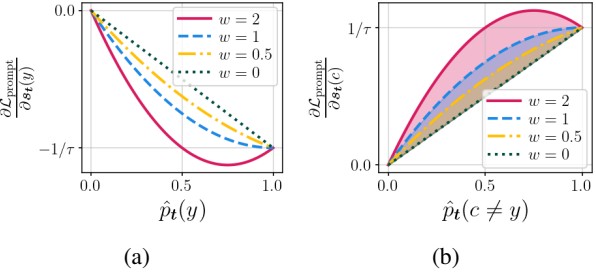

Figure 3: Gradient component of $\mathcal{L}_{\text{prompt}}$ with respect to (a) $\mathbf{s}_t(c \neq y)$ and (b) $\mathbf{s}_t(y)$, where $w = 0$ represents standard cross-entropy.

where $w$ is a hyperparameter that balances the contribution of CoA-loss.

To illustrate how CoA-loss refines the decision boundary between confusing classes, we analyze the gradients for the correct class $y$ and an incorrect class $c$. The gradients of $\mathcal{L}_{\text{prompt}}$ with respect to the similarities $\mathbf{s}_t(y)$ and $\mathbf{s}_t(c \neq y)$ are as follows:

$$\frac{\partial \mathcal{L}_{\text{prompt}}}{\partial \mathbf{s}_t(y)} = -\frac{1}{\tau}(1 - \hat{p}_t(y))(1 - w\hat{p}_t(y)),$$
$$\frac{\partial \mathcal{L}_{\text{prompt}}}{\partial \mathbf{s}_t(c \neq y)} = \frac{1}{\tau}\hat{p}_t(c)(1 + w\hat{p}_t(y)). \quad (11)$$

Figure 3 shows the gradients for classes $y$ and $c$. When $w = 0$, the loss corresponds to standard cross-entropy. As implied by Equation (11), standard cross-entropy assigns gradients based on individual probabilities, ignoring inter-class relationships. In contrast, CoA-loss increases the gradient for $y$ when $\hat{p}_t(y)$ approaches 0.5 and for class $c$ when $\hat{p}_t(c)$ nears $\hat{p}_t(y)$, especially as $w$ increases. These scenarios represent confusing cases where the correct class probabil-

ity is uncertain or where the incorrect class probability is similar to the correct class, making them difficult to distinguish. CoA-loss thus induces larger gradient updates for these confusing cases.

### 3.4. Confidence-Aware Weights for Generalization without Trade-Offs

Equation (6) suggests that optimizing $\boldsymbol{\pi}$ can minimize the upper bound of the expected error $\epsilon_T(\hat{p}_{\mathcal{T}}^{\boldsymbol{\pi}})$ of the mixture model. Achieving this requires assigning a higher weight to the prediction with lower error in the target domain. In this section, we propose confidence-aware weights (CoA-weights), which adjust the weight of each prediction based on the class domain.

**Assumption 3.4.** The specialized prediction $\hat{p}_{\boldsymbol{t}_i}$ for $\mathcal{D}_{T_i}$ satisfies the following relationships:

$$\epsilon_{T_i}(\hat{p}_{\boldsymbol{t}_i}) \leq \epsilon_{T_i}(\hat{p}_{\boldsymbol{t}_{j\neq i}}) \quad \text{and} \quad \epsilon_{T_{j\neq i}}(\hat{p}_{\boldsymbol{t}_0}) \leq \epsilon_{T_{j\neq i}}(\hat{p}_{\boldsymbol{t}_i}). \tag{12}$$

The first inequality reflects that a prediction $\hat{p}_{\boldsymbol{t}_i}$ optimized for a specific domain $\mathcal{D}_{T_i}$ always performs better than predictions $\hat{p}_{\boldsymbol{t}_{j\neq i}}$ made by prompts optimized for other domains. Conversely, the second inequality assumes that the generalized prediction $\hat{p}_{\boldsymbol{t}_0}$ is more effective for unseen classes. This assumption aligns with the principle that a single prediction specialized for a task cannot excel at all tasks, a claim further supported by statistical evidence presented in Appendix B.

Equation (6) and Assumption 3.4 suggest that minimizing the upper bound of $\epsilon_T(\hat{p}_{\mathcal{T}}^{\boldsymbol{\pi}})$ requires increasing $\pi^{\text{in}}$ for in-classes and decreasing $\pi^{\text{out}}$ for out-classes. To achieve this, we propose CoA-weights which adjusts the weights $\pi_i^{\text{in}}$ and $\pi_i^{\text{out}}$ of $\hat{p}_{\boldsymbol{t}_i}$ in the mixture model.

**Optimizing $\pi_i^{\text{in}}$ for In-Class Domains** The weight $\pi_i^{\text{in}}$ for in-class domains is optimized by minimizing the cross-entropy loss of the mixture model over the training domain $\mathcal{D}_{S_i}$:

$$\pi_i^{\text{in}} = \arg\min_{\pi_i^{\text{in}}} \mathbb{E}_{(\mathbf{x},y)\sim\mathcal{D}_{S_i}} [\mathcal{L}_{\text{CE}}(\mathbf{x}, y; \hat{p}_{\mathcal{T}}^{\boldsymbol{\pi}})]. \tag{13}$$

With this optimization, weight $\pi_i^{\text{in}}$ is increased when the specialized prediction $\hat{p}_{\boldsymbol{t}_i}$ outperforms the generalized prediction $\hat{p}_{\boldsymbol{t}_{j\neq i}}$ for the in-class domain and decreased otherwise. Consequently, the weight $\pi_i^{\text{in}}$ for in-classes is optimized. Further details on the cross-entropy effect in the mixture model are provided in Appendix C.

**Optimizing $\pi_i^{\text{out}}$ for Out-Class Domains** Because only in-classes are available during training, out-class set $\mathcal{Y}_i^{\text{out}}$ must be generated to optimize $\pi_i^{\text{out}}$. The out-class set can be

generated by combining random strings or retrieving random words. According to Assumption 3.4, the temperature $\pi_i^{\text{out}}$ for out-class domains is optimized using entropy loss $\mathcal{L}_{\text{Ent}}$, which compares the entropy of $\hat{p}_{\boldsymbol{t}_i}$ and $\hat{p}_{\boldsymbol{t}_0}$ as follows:

$$\pi_i^{\text{out}} = \arg\min_{\pi_i^{\text{out}}} \mathbb{E}_{(\mathbf{x},y)\sim\mathcal{D}_{S_i}} [\mathcal{L}_{\text{Ent}}(\mathbf{x}; \hat{p}_{\boldsymbol{t}_i}, \hat{p}_{\boldsymbol{t}_0})], \tag{14}$$

$$\mathcal{L}_{\text{Ent}} = \max\left(0, H(\hat{p}_{\boldsymbol{t}_0}) - H(\hat{p}_{\boldsymbol{t}_i}) + d\right), \tag{15}$$

where $d$ is a margin and $H(\hat{p})$ is the normalized entropy of $\hat{p}$ over the out-class set, i.e., $H(\hat{p}) = \sum_{c\sim\mathcal{Y}_i^{\text{out}}} -\hat{p}(c)\log\hat{p}(c)/\log|\mathcal{Y}_i^{\text{out}}|$.

The entropy loss $\mathcal{L}_{\text{Ent}}$ ensures that the entropy of $\hat{p}_{\boldsymbol{t}_{i>0}}$ exceeds that of $\hat{p}_{\boldsymbol{t}_0}$ by a margin of $d$. Entropy measures uncertainty in the prediction, with higher values indicating lower confidence. As a result, $\pi_i^{\text{out}}$ is optimized to make specialized predictions less confident than generalized ones.

## 4. Experiments

### 4.1. Datasets and Implementation Details

We validate the effectiveness of our method in three tasks: (1) base-to-new generalization, (2) few-shot class-incremental learning (FSCIL), and (3) cross-dataset transfer.

**Datasets** We evaluate base-to-new generalization and cross-dataset transfer performance using 11 datasets: ImageNet (Deng et al., 2009), Caltech101 (Fei-Fei et al., 2004), OxfordPets (Parkhi et al., 2012), StanfordCars (Krause et al., 2013), Flowers102 (Nilsback & Zisserman, 2008), Food101 (Bossard et al., 2014), FGVCAircraft (Maji et al., 2013), EuroSAT (Helber et al., 2019), UCF101 (Soomro, 2012), DTD (Cimpoi et al., 2014), and SUN397 (Xiao et al., 2010). For FSCIL, we use CIFAR100 (Krizhevsky et al., 2009). Following Tao et al. (2020), we split the classes into 60 *Base* and 40 *New* classes and adopted a 5-shot 5-way setting, resulting in a total of 9 training sessions.

**Training Details** The prompt length $M$ is initialized randomly and set to 16 unless specified. The out-class set $\mathcal{Y}_i^{\text{out}}$ for optimizing $\pi_i^{\text{out}}$ is generated by sampling the same number of random words as the in-class set $\mathcal{Y}_i$ using the API (Rebguns, 2021). Prompt tuning is performed using the Adam optimizer (Kingma, 2014) with a learning rate of 0.002. Optimization for the CoA-weights is conducted with SGD. Further details of implementation are provided in Appendix D.

### 4.2. Performance Comparison

**Base-to-New Generalization** We evaluate prompt tuning performance over classes in a 4-shot setting. Each dataset is evenly split into two disjoint subsets: base classes (*Base*) for tuning and unseen new classes (*New*). Accuracy is measured

Table 1: Performance comparison on 11 datasets in the base-to-new benchmark. H represents the harmonic mean.

| Method | AVERAGE | | | IMAGENET | | | CALTECH101 | | |
|---|---|---|---|---|---|---|---|---|---|
| | BASE | NEW | H | BASE | NEW | H | BASE | NEW | H |
| CLIP | 65.14 | 68.78 | 66.82 | 64.43 | 60.04 | 62.16 | 90.64 | 91.16 | 90.90 |
| CoOp | 77.23 | 68.56 | 71.33 | $73.72 \pm 0.29$ | $64.94 \pm 0.87$ | 69.05 | $97.16 \pm 0.16$ | $93.92 \pm 0.80$ | 95.51 |
| ProGrad | 78.74 | 72.19 | 75.06 | $74.81 \pm 0.29$ | $66.68 \pm 0.26$ | 70.51 | $97.50 \pm 0.08$ | $95.49 \pm 0.27$ | 96.48 |
| KgCoOp | 78.67 | 74.62 | 76.38 | $75.44 \pm 0.08$ | $69.43 \pm 0.29$ | 72.31 | $97.61 \pm 0.33$ | $94.80 \pm 0.45$ | 96.18 |
| MaPLe | 77.14 | 72.91 | 74.69 | $75.40 \pm 0.29$ | $\mathbf{70.43 \pm 0.12}$ | $\mathbf{72.83}$ | $97.47 \pm 0.31$ | $93.77 \pm 1.11$ | 95.57 |
| DePT | 79.20 | 66.36 | 71.78 | $73.50 \pm 0.22$ | $70.00 \pm 0.16$ | 71.71 | $97.83 \pm 0.05$ | $\mathbf{95.83 \pm 0.25}$ | 96.82 |
| CoA-loss | 79.12 | 73.66 | 76.15 | $\mathbf{75.68 \pm 0.00}$ | $67.98 \pm 0.31$ | 71.62 | $97.94 \pm 0.14$ | $94.54 \pm 0.24$ | 96.21 |
| CoCoA-Mix | $\mathbf{79.31}$ | $\mathbf{75.10}$ | $\mathbf{77.03}$ | $75.47 \pm 0.09$ | $68.92 \pm 0.10$ | 72.04 | $\mathbf{98.02 \pm 0.03}$ | $94.39 \pm 0.10$ | 96.17 |

| Method | OXFORDPETS | | | STANFORDCARS | | | FLOWERS102 | | |
|---|---|---|---|---|---|---|---|---|---|
| | BASE | NEW | H | BASE | NEW | H | BASE | NEW | H |
| CLIP | 90.01 | 94.24 | 92.07 | 55.37 | 66.65 | 60.49 | 69.23 | 73.90 | 71.49 |
| CoOp | $94.10 \pm 0.73$ | $94.42 \pm 4.17$ | 94.16 | $69.54 \pm 0.75$ | $71.39 \pm 1.28$ | 70.44 | $90.60 \pm 1.50$ | $67.00 \pm 1.04$ | 77.01 |
| ProGrad | $95.00 \pm 0.31$ | $97.36 \pm 0.42$ | 96.16 | $71.45 \pm 0.39$ | $73.16 \pm 0.58$ | 72.29 | $91.36 \pm 0.63$ | $74.92 \pm 0.90$ | 82.32 |
| KgCoOp | $94.65 \pm 0.15$ | $97.59 \pm 0.08$ | 96.10 | $68.64 \pm 0.35$ | $74.96 \pm 0.53$ | 71.66 | $90.09 \pm 0.63$ | $76.31 \pm 0.42$ | 82.63 |
| MaPLe | $94.80 \pm 0.94$ | $97.67 \pm 0.21$ | 96.21 | $67.97 \pm 0.29$ | $74.40 \pm 0.45$ | 71.04 | $88.03 \pm 1.62$ | $73.43 \pm 0.49$ | 80.06 |
| DePT | $94.00 \pm 0.29$ | $88.63 \pm 0.78$ | 91.23 | $71.83 \pm 0.52$ | $59.27 \pm 0.76$ | 64.94 | $\mathbf{94.53 \pm 0.53}$ | $66.30 \pm 1.42$ | 77.92 |
| CoA-loss | $94.90 \pm 0.49$ | $\mathbf{97.93 \pm 0.08}$ | $\mathbf{96.39}$ | $72.70 \pm 0.11$ | $73.07 \pm 1.27$ | 72.87 | $88.89 \pm 1.75$ | $75.58 \pm 1.31$ | 81.67 |
| CoCoA-Mix | $\mathbf{95.16 \pm 0.38}$ | $97.60 \pm 0.09$ | 96.36 | $\mathbf{73.09 \pm 0.25}$ | $74.97 \pm 0.08$ | $\mathbf{74.01}$ | $91.04 \pm 1.79$ | $\mathbf{77.37 \pm 0.38}$ | 83.64 |

| Method | FOOD101 | | | FGVCAIRCRAFT | | | SUN397 | | |
|---|---|---|---|---|---|---|---|---|---|
| | BASE | NEW | H | BASE | NEW | H | BASE | NEW | H |
| CLIP | 83.58 | 84.95 | 84.26 | 19.51 | 24.60 | 21.76 | 66.76 | 70.52 | 68.59 |
| CoOp | $89.19 \pm 0.19$ | $88.45 \pm 0.89$ | 88.81 | $26.17 \pm 7.89$ | $19.50 \pm 11.94$ | 11.46 | $77.37 \pm 0.66$ | $72.06 \pm 1.56$ | 74.60 |
| ProGrad | $89.33 \pm 0.08$ | $89.93 \pm 0.58$ | 89.63 | $34.21 \pm 1.99$ | $28.53 \pm 2.08$ | 30.97 | $\mathbf{79.16 \pm 0.36}$ | $74.34 \pm 0.75$ | 76.20 |
| KgCoOp | $\mathbf{90.26 \pm 0.11}$ | $\mathbf{91.25 \pm 0.15}$ | $\mathbf{90.75}$ | $33.43 \pm 0.56$ | $32.27 \pm 1.19$ | 32.81 | $79.07 \pm 0.24$ | $76.78 \pm 0.24$ | 77.91 |
| MaPLe | $89.37 \pm 0.54$ | $90.77 \pm 0.54$ | 90.06 | $31.67 \pm 0.66$ | $33.13 \pm 2.38$ | 32.29 | $78.33 \pm 0.21$ | $\mathbf{77.67 \pm 0.45}$ | $\mathbf{78.00}$ |
| DePT | $89.80 \pm 0.08$ | $88.10 \pm 0.16$ | 88.94 | $\mathbf{35.93 \pm 0.93}$ | $24.33 \pm 0.09$ | 29.01 | $79.10 \pm 0.22$ | $67.27 \pm 0.46$ | 72.70 |
| CoA-loss | $90.11 \pm 0.18$ | $90.87 \pm 0.42$ | 90.49 | $33.91 \pm 0.68$ | $32.47 \pm 0.37$ | 33.17 | $78.70 \pm 0.25$ | $75.43 \pm 0.72$ | 77.03 |
| CoCoA-Mix | $90.09 \pm 0.16$ | $90.93 \pm 0.09$ | 90.50 | $33.51 \pm 0.28$ | $\mathbf{34.15 \pm 0.14}$ | $\mathbf{33.83}$ | $78.51 \pm 0.17$ | $76.60 \pm 0.24$ | 77.54 |

| Method | DTD | | | EuroSAT | | | UCF101 | | |
|---|---|---|---|---|---|---|---|---|---|
| | BASE | NEW | H | BASE | NEW | H | BASE | NEW | H |
| CLIP | 53.24 | 54.71 | 53.97 | 54.79 | 66.21 | 59.96 | 69.03 | 69.61 | 69.32 |
| CoOp | $71.22 \pm 1.13$ | $53.62 \pm 3.45$ | 61.03 | $79.93 \pm 1.07$ | $64.79 \pm 6.36$ | 71.19 | $80.58 \pm 0.66$ | $64.11 \pm 2.84$ | 71.32 |
| ProGrad | $72.07 \pm 0.29$ | $50.56 \pm 2.43$ | 59.35 | $81.29 \pm 3.36$ | $69.81 \pm 5.56$ | 74.80 | $80.97 \pm 0.29$ | $73.32 \pm 1.85$ | 76.93 |
| KgCoOp | $72.92 \pm 1.05$ | $59.14 \pm 1.53$ | 65.28 | $83.20 \pm 0.72$ | $70.51 \pm 9.30$ | 75.61 | $80.09 \pm 0.24$ | $\mathbf{77.75 \pm 0.40}$ | 78.90 |
| MaPLe | $70.40 \pm 2.57$ | $58.40 \pm 3.00$ | 63.71 | $76.50 \pm 3.85$ | $55.70 \pm 3.19$ | 64.27 | $78.57 \pm 2.11$ | $76.60 \pm 1.56$ | 77.53 |
| DePT | $\mathbf{74.40 \pm 0.83}$ | $53.13 \pm 1.07$ | 61.98 | $78.70 \pm 1.56$ | $50.53 \pm 5.71$ | 61.08 | $\mathbf{81.57 \pm 0.84}$ | $66.53 \pm 0.87$ | 73.28 |
| CoA-loss | $73.23 \pm 2.02$ | $58.09 \pm 0.81$ | 64.76 | $83.38 \pm 0.49$ | $70.07 \pm 2.49$ | 76.09 | $80.83 \pm 0.80$ | $74.22 \pm 0.91$ | 77.38 |
| CoCoA-Mix | $72.80 \pm 1.89$ | $\mathbf{64.29 \pm 1.25}$ | $\mathbf{68.25}$ | $83.49 \pm 0.66$ | $69.11 \pm 3.10$ | 75.54 | $81.28 \pm 0.95$ | $\mathbf{77.75 \pm 0.24}$ | 79.47 |

independently on *Base* and *New*, and their harmonic mean H (Xian et al., 2017) is calculated to evaluate the trade-off between them. The final performance is reported as the average of three random seeds for a fair evaluation.

Table 1 shows that CoOp (Zhou et al., 2022c) achieves a higher average performance than CLIP on *Base* but a lower average performance on *New*, highlighting the need for generalization in prompt tuning. ProGrad (Zhu et al., 2023) and KgCoOp (Yao et al., 2023), which incorporate hand-crafted prompts during training, improve average performance on both *Base* and *New*. MaPLe (Khattak et al., 2023a) and DePT (Zhang et al., 2024b) focus on improving performance on both *Base* and *New* by increasing model capacity. However, with only 4-shot training samples, they often overfit, limiting generalization. Our CoA-loss, when combined with a naive ensemble, improves *Base* performance but offers limited generalization as it only focuses on specialization. By incorporating CoA-weights, our CoCoA-Mix achieves the highest average performance on both *Base* and *New* without trade-offs. Notably, CoCoA-Mix surpasses the specialization performance of DePT while using 2.8% of its parameters.

Table 2: Performance comparison on CIFAR100 in the FSCIL benchmark. Mean represents the average accuracy across all sessions, and PD indicates the performance difference between the first and last sessions.

| METHOD | ACC(%)↑ | | | | | | | | | MEAN↑ | PD↓ |
|---|---|---|---|---|---|---|---|---|---|---|---|
| | 0 | 1 | 2 | 3 | 4 | 5 | 6 | 7 | 8 | | |
| L2P | **89.9** | **86.0** | 81.8 | 80.3 | 80.0 | 74.6 | 73.2 | 72.6 | 65.0 | 78.2 | 24.9 |
| CLIP-ZSL | – | – | – | – | – | – | – | – | – | 77.9 | – |
| CoOp-FSCIL | 88.6 | 78.9 | 77.5 | 76.0 | 76.8 | 78.3 | 79.2 | 79.8 | 79.3 | 79.4 | 9.3 |
| FACT w/ CLIP | 87.8 | 84.0 | 81.4 | 78.0 | 77.8 | 76.3 | 75.0 | 72.5 | 71.9 | 78.3 | 15.9 |
| FSPT-FSCIL | 86.9 | 83.1 | 81.9 | 80.7 | 80.4 | 79.9 | 80.1 | 79.9 | 79.4 | 81.4 | 7.5 |
| CoCoA-Mix (Ours) | 88.2 | 85.6 | **84.6** | **82.7** | **82.8** | **82.5** | **82.3** | **81.8** | **80.8** | **83.5** | **7.4** |

Table 3: Performance comparison in cross-dataset transfer

| METHOD | SOURCE | TARGET | H |
|---|---|---|---|
| CLIP | 66.73 | 64.89 | 63.97 |
| CoOp | 69.06 ± 0.43 | 59.88 | 61.52 |
| ProGrad | 70.21 ± 0.16 | 62.36 | 63.58 |
| KgCoOp | 70.52 ± 0.05 | 64.45 | 65.17 |
| MaPLe | 69.53 ± 0.39 | 65.24 | 65.26 |
| DePT | 68.03 ± 0.09 | 65.06 | 64.42 |
| CoCoA-Mix | **70.85 ± 0.09** | **65.27** | **66.07** |

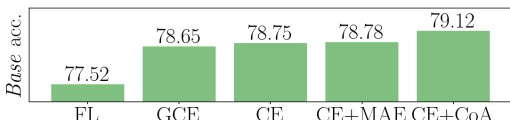

Figure 4: Comparison of *Base* performance by loss function

**Few-Shot Class-Incremental Learning (FSCIL)** We evaluated our method on FSCIL tasks to verify its effectiveness in incremental learning with limited data. In FSCIL, the number of prompts $K$ was increased incrementally, with each prompt specializing in its session. For a fair comparison, CoCoA-Mix uses prompts with $M = 2$ per session. It has fewer parameters than methods with $M = 16$ until session 6 but requires more parameters from session 7. Table 2 shows the performance of FSCIL on the CIFAR100 dataset. L2P (Wang et al., 2022b), which dynamically selects prompts from a pool, shows high initial performance but suffers significant knowledge forgetting as new classes are added, leading to the highest performance difference (PD). CoOp-FSCIL (Zhou et al., 2022c) and FACT w/ CLIP (Zhou et al., 2022a) outperform zero-shot CLIP in early sessions but are affected by knowledge forgetting in later sessions. FSPT-FSCIL (Ran et al., 2024) outperforms zero-shot CLIP in all sessions by leveraging a brain-inspired strategy in prompt tuning. CoCoA-Mix performs lower in the first two sessions due to fewer parameters but achieves the highest performance in later sessions, outperforming the state-of-the-art methods. Details of the FSCIL implementation are provided in Appendix D.4.

**Cross-Dataset Transfer** We evaluate cross-dataset transfer by training on ImageNet with $1,000$ classes in a 4-shot setting and testing on 10 different datasets. Table 3 shows the accuracy and harmonic mean for source and target datasets. CoOp improves source accuracy over zero-shot CLIP but

reduces target accuracy, highlighting the need for generalization. ProGrad and KgCoOp similarly enhance source accuracy but fail to outperform zero-shot CLIP on the target dataset, indicating limited transferability of prompt knowledge. MaPLe and DePT improve transfer performance by introducing additional parameters but show limited source accuracy. CoCoA-Mix achieves the highest performance on both source and target datasets, demonstrating effective knowledge transfer across different datasets. Notably, CoCoA-Mix outperforms existing methods in generalization while utilizing only 0.26% of the parameters used by MaPLe. Appendix E.1 provides detailed results on 10 datasets.

### 4.3. Effectiveness of Our Method

**Performance Comparison with Existing Loss Functions** We compared *Base* performance across various loss functions to demonstrate the effectiveness of CoA-loss for specialization in prompt tuning. Figure 4 shows the average *Base* performance for each loss function. Focal loss (FL) (Ross & Dollár, 2017) emphasizes misclassified samples but struggles to learn from well-classified ones, leading to poor performance compared with CE in prompt tuning. Generalized cross entropy (GCE) (Zhang & Sabuncu, 2018), which generalizes CE and mean absolute error (MAE), limits optimization on misclassified samples, restricting specialization. CE achieves better performance in prompt tuning by balancing well-classified and misclassified samples. MAE combined with CE treats all samples more equally and ensures consistent learning even in the few-shot setting, slightly improving performance. However, these loss functions fail to address confusing cases that significantly affect decision boundaries, limiting their performance. Adding

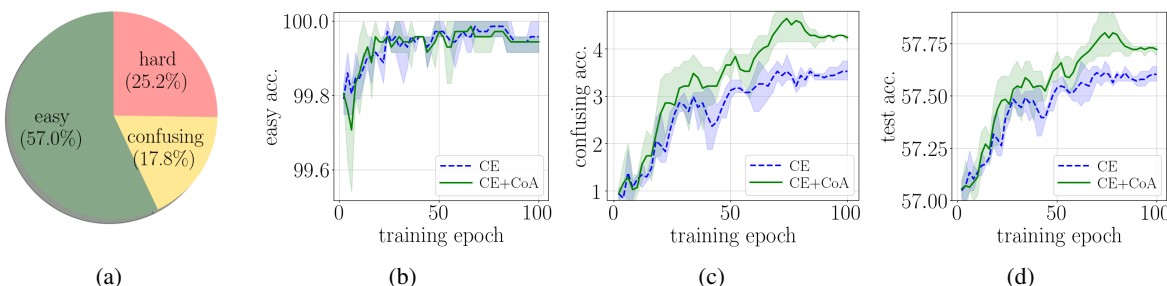

(a)        (b)        (c)        (d)

Figure 5: (a) Proportion of predictions by zero-shot CLIP on EuroSAT. (b) Accuracy on easy test samples correctly predicted by zero-shot CLIP. (c) Accuracy on confusing test samples misclassified by zero-shot CLIP with a probability gap below 0.2. (d) Accuracy on all test samples.

Table 4: Comparison of performance based on the loss function used to train prompts and the strategy of mixing prompts. $T_{En}$ refers to the tuning ensemble methods (Lu et al., 2024). ViT-B/16 shows the average performance over 11 datasets, while RN50+RN101+ViT-B/16+ViT-B/32 reports the average over 10 datasets.

| Backbone | Method | Loss | Ensemble | Base | *New* | H |
|---|---|---|---|---|---|---|
| | CLIP | - | - | 65.1 | 68.8 | 66.8 |
| | CLIP (w/ Ensemble) | - | Uniform Ensemble | 70.6 | 74.3 | 72.3 |
| ViT-B/16 | CoA-loss (w/o Ensemble) | CoA-loss | - | 78.6 | 68.5 | 72.9 |
| | CoA-loss (w/ Ensemble) | CoA-loss | Uniform Ensemble | 79.1 | 73.7 | 76.2 |
| | CoCoA-Mix | CoA-loss | CoA-weights | **79.3** | **75.1** | **77.0** |
| RN50 + RN101 + ViT-B/16 + ViT-B/32 | $T_{En}$ + CoCoOp | CoCoOp | Sample-Aware Weight Generator | 84.1 | 75.5 | 79.2 |
| | $T_{En}$ + CoA-Loss | CoA-loss | Sample-Aware Weight Generator | 85.3 | 75.2 | 79.5 |
| | CoCoA-Mix | CoA-loss | CoA-weights | **85.4** | **76.3** | **80.3** |

CoA-loss to CE achieves the best performance by effectively focusing on confusing cases. Detailed results and analysis are provided in Appendix E.3.

**Effectiveness of Confusion-Aware Loss** We analyze the effect of CoA-loss on model predictions using the EuroSAT dataset. Figure 5 compares training progress with and without CoA-loss. Figure 5(a) shows the proportion of predictions by zero-shot CLIP. Easy samples are correctly classified, while confusing samples are misclassified with a probability gap of less than 0.2 between the correct and incorrect classes. Hard samples are misclassified with a larger probability gap. Figure 5(b) and Figure 5(c) present performance on easy samples and confusing samples. While CoA-loss performs similarly to CE on easy samples, it significantly improves accuracy on confusing ones. Figure 5(d) shows that CoA-loss outperforms CE on test data by addressing confusing cases, enhancing class decision boundaries. The performance improvement on confusing samples with CoA-loss for each dataset is provided in Appendix E.2

**Effectiveness of Confidence-Aware Weights** We evaluate the effectiveness of CoA-weights in the context of prompt ensembling. As shown in Table 4, we compare var-

ious ensembling strategies and loss functions across both single- and multi-backbone configurations. In the ViT-B/16 setting, uniform ensembling improves generalization performance over the CLIP baseline by aggregating predictions from multiple prompts. CoCoA-Mix, which integrates CoA-loss and CoA-weights, outperforms all other variants. This highlights the importance of confidence-aware weighting. In the model-level ensemble setting, CoCoA-Mix achieves the best performance, outperforming the tuning ensemble ($T_{En}$) with a sample-aware weight generator (Lu et al., 2024). Notably, CoA-weights achieve this with only two learnable parameters, compared with over 205,204 parameters required by the sample-aware weight generator.

## Conclusion

We proposed CoCoA-Mix, a framework combining CoA-loss and CoA-weights to enhance specialization and generalization in prompt tuning. CoA-loss improves specialization by addressing confusing cases, while CoA-weights adjust the confidence of predictions to enhance generalization without sacrificing specialization. We believe our method sets a new direction for effective prompt tuning.

## Acknowledgements

This work was supported by Korea Evaluation Institute Of Industrial Technology (KEIT) grant funded by the Korea government(MOTIE) (No.20023455, Development of Cooperate Mapping, Environment Recognition and Autonomous Driving Technology for Multi Mobile Robots Operating in Large-scale Indoor Workspace). This research was supported by the KAIST Convergence Research Institute Operation Program.

## Impact Statement

Prompt tuning has emerged as a promising method for VLMs, enabling efficient adaptation to diverse tasks by optimizing a small set of parameters while keeping the core model frozen. However, real-world scenarios with unseen classes require more generalized solutions. We proposed a mixture model-based approach to enhance generality and improve performance in practical applications. We believe the outcomes of this research are largely positive, making it unnecessary to highlight specific negative impacts in this paper.

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

## A. Proof of Theorem 3.2

Consider $K + 1$ individual prompts $\mathcal{T} = \{\boldsymbol{t}_0, \boldsymbol{t}_1, \ldots, \boldsymbol{t}_K\}$ and a mixture model $\hat{p}_{\mathcal{T}}^{\boldsymbol{\pi}}$ with non-negative weights $\boldsymbol{\pi} = \{\pi_0, \pi_1, \ldots, \pi_K\}$, where $\sum_{i=0}^{K} \pi_i = 1$. Let $\mathcal{D}_T$ be an arbitrary target domain. The expected error $\epsilon_T(\hat{p}_{\mathcal{T}}^{\boldsymbol{\pi}})$ of the mixture model on the target domain is defined as follows in terms of the Kullback-Leibler (KL) divergence:

$$\epsilon_T(\hat{p}_{\mathcal{T}}^{\boldsymbol{\pi}}) = \mathbb{E}_{(\mathbf{x},y)\sim\mathcal{D}_T}\left[-\log \hat{p}_{\mathcal{T}}^{\boldsymbol{\pi}}(y)\right],$$

where $y$ is the ground-truth label for the image $\mathbf{x}$.

Using the definition of the mixture model, $\hat{p}_{\mathcal{T}}^{\boldsymbol{\pi}}(y) = \frac{\exp\left(\sum_{i=0}^{K} \pi_i \boldsymbol{s}_{\boldsymbol{t}_i}(y)/\tau\right)}{\sum_{l'\in\mathcal{Y}} \exp\left(\sum_{i=0}^{K} \pi_i \boldsymbol{s}_{\boldsymbol{t}_i}(l')/\tau\right)}$, the expected error can be decomposed into two terms as follows:

$$\begin{aligned}
\epsilon_T(\hat{p}_{\mathcal{T}}^{\boldsymbol{\pi}}) &= \mathbb{E}_{(\mathbf{x},y)\sim\mathcal{D}_T}\left[-\log \hat{p}_{\mathcal{T}}^{\boldsymbol{\pi}}(y)\right] \\
&= \mathbb{E}_{(\mathbf{x},y)\sim\mathcal{D}_T}\left[-\log \frac{\exp\left(\sum_{i=0}^{K} \pi_i \boldsymbol{s}_{\boldsymbol{t}_i}(y)/\tau\right)}{\sum_{l'\in\mathcal{Y}} \exp\left(\sum_{i=0}^{K} \pi_i \boldsymbol{s}_{\boldsymbol{t}_i}(l')/\tau\right)}\right] \\
&= \mathbb{E}_{(\mathbf{x},y)\sim\mathcal{D}_T}\left[-\sum_{i=0}^{K} \pi_i \boldsymbol{s}_{\boldsymbol{t}_i}(y)/\tau + \log \sum_{l'\in\mathcal{Y}} \exp\left(\sum_{i=0}^{K} \pi_i \boldsymbol{s}_{\boldsymbol{t}_i}(l')/\tau\right)\right] \\
&= \mathbb{E}_{(\mathbf{x},y)\sim\mathcal{D}_T}\left[-\sum_{i=0}^{K} \pi_i \boldsymbol{s}_{\boldsymbol{t}_i}(y)/\tau + \sum_{i=0}^{K} \pi_i \log \sum_{l'\in\mathcal{Y}} \exp\left(\boldsymbol{s}_{\boldsymbol{t}_i}(l')/\tau\right)\right] \\
&\quad + \mathbb{E}_{(\mathbf{x},y)\sim\mathcal{D}_T}\left[-\sum_{i=0}^{K} \pi_i \log \sum_{l'\in\mathcal{Y}} \exp\left(\boldsymbol{s}_{\boldsymbol{t}_i}(l')/\tau\right) + \log \sum_{l'\in\mathcal{Y}} \exp\left(\sum_{i=0}^{K} \pi_i \boldsymbol{s}_{\boldsymbol{t}_i}(l')/\tau\right)\right].
\end{aligned}$$

The first term is rewritten using the definition of the individual predictive distribution $\hat{p}_{\boldsymbol{t}_i}$ for the prompt $\boldsymbol{t}_i$, given as $\hat{p}_{\boldsymbol{t}_i}(y) = \frac{\exp\left(\boldsymbol{s}_{\boldsymbol{t}_i}(y)/\tau\right)}{\sum_{l'\in\mathcal{Y}} \exp\left(\boldsymbol{s}_{\boldsymbol{t}_i}(l')/\tau\right)}$, as follows:

$$\begin{aligned}
&\mathbb{E}_{(\mathbf{x},y)\sim\mathcal{D}_T}\left[-\sum_{i=0}^{K} \pi_i \boldsymbol{s}_{\boldsymbol{t}_i}(y)/\tau + \sum_{i=0}^{K} \pi_i \log \sum_{l'\in\mathcal{Y}} \exp\left(\boldsymbol{s}_{\boldsymbol{t}_i}(l')/\tau\right)\right] \\
&= \mathbb{E}_{(\mathbf{x},y)\sim\mathcal{D}_T}\left[-\sum_{i=0}^{K} \pi_i \left(\boldsymbol{s}_{\boldsymbol{t}_i}(y)/\tau - \log \sum_{l'\in\mathcal{Y}} \exp\left(\boldsymbol{s}_{\boldsymbol{t}_i}(l')/\tau\right)\right)\right] \\
&= \mathbb{E}_{(\mathbf{x},y)\sim\mathcal{D}_T}\left[-\sum_{i=0}^{K} \pi_i \left(\log \exp\left(\boldsymbol{s}_{\boldsymbol{t}_i}(y)/\tau\right) - \log \sum_{l'\in\mathcal{Y}} \exp\left(\boldsymbol{s}_{\boldsymbol{t}_i}(l')/\tau\right)\right)\right] \\
&= \sum_{i=0}^{K} \pi_i \mathbb{E}_{(\mathbf{x},y)\sim\mathcal{D}_T}\left[-\log \frac{\exp\left(\boldsymbol{s}_{\boldsymbol{t}_i}(y)/\tau\right)}{\sum_{l'\in\mathcal{Y}} \exp\left(\boldsymbol{s}_{\boldsymbol{t}_i}(l')/\tau\right)}\right] \\
&= \sum_{i=0}^{K} \pi_i \epsilon_T(\hat{p}_{\boldsymbol{t}_i}).
\end{aligned}$$

As a result, the first term is equivalent to a convex combination of the expected errors of the individual predictive distributions with weights $\boldsymbol{\pi}$.

For the second term, Jensen's inequality (Jensen, 1906) can be applied to bound it, as $\log \sum \exp$ is a convex function:

$$\mathbb{E}_{(\mathbf{x},y)\sim\mathcal{D}_T}\left[-\sum_{i=0}^{K}\pi_i\log\sum_{l'\in\mathcal{Y}}\exp\left(\boldsymbol{s}_{\boldsymbol{t}_i}(l')/\tau\right)+\log\sum_{l'\in\mathcal{Y}}\exp\left(\sum_{i=0}^{K}\pi_i\boldsymbol{s}_{\boldsymbol{t}_i}(l')/\tau\right)\right]$$

$$\leq\mathbb{E}_{(\mathbf{x},y)\sim\mathcal{D}_T}\left[-\sum_{i=0}^{K}\pi_i\log\sum_{l'\in\mathcal{Y}}\exp\left(\boldsymbol{s}_{\boldsymbol{t}_i}(l')/\tau\right)+\sum_{i=0}^{K}\pi_i\left(\log\sum_{l'\in\mathcal{Y}}\exp\left(\boldsymbol{s}_{\boldsymbol{t}_i}(l')\tau\right)\right)\right]$$

$$\leq 0.$$

By combining the results from the first and second terms, we conclude that the expected error of the mixture model on the target domain is bounded as follows:

$$\epsilon_T(\hat{p}_{\mathcal{T}}^{\boldsymbol{\pi}}) \leq \sum_i \pi_i\epsilon_T(\hat{p}_{\boldsymbol{t}_i}).$$

## B. Statistical Validation of Assumption 3.4

To empirically verify Assumption 3.4, we conducted a statistical experiment using the CIFAR-100 (Krizhevsky et al., 2009) dataset and the CLIP model. Specifically, we randomly partitioned the 100 classes into 50 in-classes and 50 out-classes. We then trained specialized prompts $\boldsymbol{t}_i$ on the in-class subset using prompt tuning and compared their predictions against the zero-shot CLIP baseline on both domains. This process was repeated over 10 random splits.

Figure 6 presents a box plot summarizing the performance differences between the specialized prompt $\boldsymbol{t}_i$ on the in-class domain $\mathcal{D}_{T_i}$ and the generalized zero-shot prompt $\boldsymbol{t}_0$. The results show that $\boldsymbol{t}_i$ consistently outperforms $\boldsymbol{t}_0$ on in-class samples from $\mathcal{D}_{T_i}$, whereas $\boldsymbol{t}_0$ achieves higher accuracy on out-class samples from $\mathcal{D}_{T_{j\neq i}}$.

To assess statistical significance, we performed one-sided paired $t$-tests on the per-split accuracy gaps. The resulting $p$-values were $9.25\times10^{-12}$ for the in-class domain comparison ($\mathrm{acc}_{\boldsymbol{t}_i} > \mathrm{acc}_{\boldsymbol{t}_0}$) and $2.06\times10^{-10}$ for the out-class domain comparison ($\mathrm{acc}_{\boldsymbol{t}_0} > \mathrm{acc}_{\boldsymbol{t}_i}$), both significantly below the standard threshold of $0.05$. These results allow us to reject the null hypothesis and confirm that both inequalities in Assumption 3.4 hold with strong statistical confidence.

These findings support the assumption that specialized predictions are more effective within their domain, whereas generalized predictions are preferable for unseen class domains.

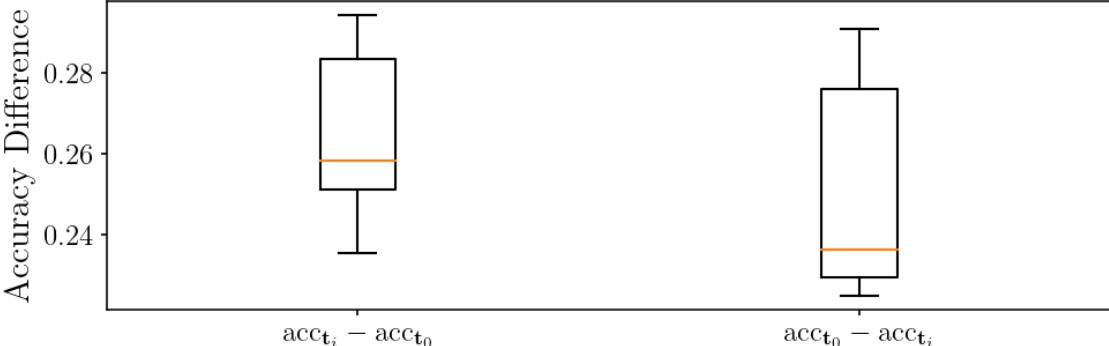

Figure 6: Box plots of accuracy differences across 10 random splits. The left shows performance gains of specialized over generalized predictions on in-class domain $\mathcal{D}_{T_i}$, i.e. $\mathrm{acc}_{\boldsymbol{t}_i} - \mathrm{acc}_{\boldsymbol{t}_0}$. The right shows gains of the generalized prediction on out-class domain $\mathcal{D}_{T_{j\neq i}}$, i.e. $\mathrm{acc}_{\boldsymbol{t}_0} - \mathrm{acc}_{\boldsymbol{t}_i}$.

## C. Effect of Cross-Entropy in the Mixture Model

The derivative of the cross-entropy $\mathcal{L}_{\text{CE}}$ for the mixture model $\hat{p}_{\mathcal{T}}^{\boldsymbol{\pi}}$ with respect to $\pi_i^{\text{in}}$ is as follows:

$$
\begin{aligned}
\frac{\partial \mathcal{L}_{\text{CE}}(\mathbf{x}, y; \hat{p}_{\mathcal{T}}^{\boldsymbol{\pi}})}{\partial \pi_i} &= \frac{\partial \left(-\log \hat{p}_{\mathcal{T}}^{\boldsymbol{\pi}}(y)\right)}{\partial \pi_i} \\
&= \frac{-1}{\hat{p}_{\mathcal{T}}^{\boldsymbol{\pi}}(y)} \frac{\partial \hat{p}_{\mathcal{T}}^{\boldsymbol{\pi}}(y)}{\partial \pi_i} \\
&= \frac{-1}{\hat{p}_{\mathcal{T}}^{\boldsymbol{\pi}}(y)} \frac{\partial}{\partial \pi_i} \left( \frac{\exp \left(\sum_{i=0}^{K} \pi_i \boldsymbol{s}_{\boldsymbol{t}_i}(y)/\tau\right)}{\sum_{l' \in \mathcal{Y}} \exp \left(\sum_{i=0}^{K} \pi_i \boldsymbol{s}_{\boldsymbol{t}_i}(l')/\tau\right)} \right) \\
&= \frac{-1}{\hat{p}_{\mathcal{T}}^{\boldsymbol{\pi}}(y)} \left( \boldsymbol{s}_{\boldsymbol{t}_i}(y)\hat{p}_{\mathcal{T}}^{\boldsymbol{\pi}}(y) - \hat{p}_{\mathcal{T}}^{\boldsymbol{\pi}}(y) \sum_{l \in \mathcal{Y}} \hat{p}_{\mathcal{T}}^{\boldsymbol{\pi}}(l)\boldsymbol{s}_{\boldsymbol{t}_i}(l) \right) / \tau \\
&= -\left( \boldsymbol{s}_{\boldsymbol{t}_i}(y) - \sum_{l \in \mathcal{Y}} \hat{p}_{\mathcal{T}}^{\boldsymbol{\pi}}(l)\boldsymbol{s}_{\boldsymbol{t}_i}(l) \right) / \tau \\
&= -\left( \boldsymbol{s}_{\boldsymbol{t}_i}(y) - \mathring{\boldsymbol{s}}_{\boldsymbol{t}_i} \right) / \tau,
\end{aligned}
$$

where $\mathring{\boldsymbol{s}}_{\boldsymbol{t}_i}$ is the importance-weighted similarity defined as a weighted sum of the predicted probability of the mixture model and the similarity derived from the prompt $\boldsymbol{t}_i$, i.e. $\mathring{\boldsymbol{s}}_{\boldsymbol{t}_i} = \sum_{l \in \mathcal{Y}} \hat{p}_{\mathcal{T}}^{\boldsymbol{\pi}}(l)\boldsymbol{s}_{\boldsymbol{t}_i}(l)$. For example, if the mixture model predicts class $l^*$ with the highest probability, $\mathring{\boldsymbol{s}}_{\boldsymbol{t}_i}$ approximates the similarity $\boldsymbol{s}_{\boldsymbol{t}_i}(l^*)$ for class $l^*$ derived from prompt $\boldsymbol{t}_i$. Here, we explain how the CoA-weights $\pi_i$ for in-classes is optimized through the cross-entropy of the mixture model. For simplicity, we assume $\mathring{\boldsymbol{s}}_{\boldsymbol{t}_i} \approx \boldsymbol{s}_{\boldsymbol{t}_i}(l^*)$, where $l^* = \arg\max_l \hat{p}_{\mathcal{T}}^{\boldsymbol{\pi}}(l)$.

In the case $\boldsymbol{s}_{\boldsymbol{t}_i}(y) > \mathring{\boldsymbol{s}}_{\boldsymbol{t}_i}$, the prompt $\boldsymbol{t}_i$ predicts the correct class $y$ with high similarity. Therefore, when the mixture model misclassifies, i.e., $l^* \neq y$, the other prompts $\boldsymbol{t}_{j \neq i}$ provide low similarities for the correct class $y$. This case results in an increase in $\pi_i$ through gradient updates, encouraging the mixture model to rely more on $\boldsymbol{t}_i$.

Conversely, if $\boldsymbol{s}_{\boldsymbol{t}_i}(y) < \mathring{\boldsymbol{s}}_{\boldsymbol{t}_i}$, the prompt $\boldsymbol{t}_i$ predicts the correct class $y$ with low similarity. When the mixture model correctly classifies, i.e. $l^* = y$, it suggests that the other prompts $\boldsymbol{t}_{j \neq i}$ provide high similarities for the correct class $y$, while the prompt $\boldsymbol{t}_i$ underperforms. This case decreases $\pi_i$, allowing the mixture model to trust the other prompts $\boldsymbol{t}_{j \neq i}$ more.

# D. Implementation Details

## D.1. Details of CoA-Weights Optimization

There exist multiple strategies for optimizing the CoA-weights $\boldsymbol{\pi}$. To evaluate the effectiveness of different optimization strategies for CoA-weights, we compared two implementation approaches: one-stage optimization and two-stage optimization. Specifically, for one-stage optimization, we design the optimization process such that the temperature scale is weighted by $\pi_i$, i.e., $\tau_i = \pi_i^{-1}\tau$. We fix $\tau_0$ to the temperature of the pre-trained CLIP model, i.e. $\tau_0 = 0.01$, and optimize $\tau_1$ jointly with prompt parameters. This enables the computation of $\boldsymbol{\pi}$ satisfying $\sum_{i=0}^{K} \pi_i = 1$, ensuring compatibility with the standard temperature scaling framework of CLIP. The optimization of a single scalar $\tau_1$ suffices to determine both $\boldsymbol{\pi}$ and the global temperature $\tau$, via the relation $\tau_i = \pi_i^{-1}\tau$. However, the temperature scale $\tau$ is not fixed during optimization, which may introduce instability when $K \geq 2$. In contrast, the two-stage strategy decouples prompt tuning and CoA-weights optimization. Here, $\boldsymbol{\pi}$ is parameterized using a softmax of parameters $\alpha_0, \cdots, \alpha_K$. Concretely, we fix the pre-softmax logit $\alpha_0 = 0$ and optimize $\alpha_i$ under a fixed temperature $\tau = 0.01$. The comparison of these strategies for the $K = 1$ case is reported in Table 5. The two strategies yield comparable performance, suggesting that the proposed loss function is robust to the specific choice of optimization parameterization for CoA-weights. In this paper, we used the one-stage strategy for most benchmarks and adopted the two-stage approach specifically for FSCIL tasks due to its stability advantages.

Table 5: Comparison of various optimization strategies with respect to $\pi_1$ and $\tau$ for *Base* and *New* domains. In one-stage optimization, $\tau_0$ is set to the temperature scale from pre-trained CLIP, i.e. 0.01. $\boldsymbol{\pi} = \{\pi_0, \pi_1\}$ and $\tau$ are determined by optimizing $\tau_1$. In two-stage optimization, $\alpha_0$ is set to 0, and $\boldsymbol{\pi}$ and $\tau$ are determined by optimizing $\alpha_1$.

| | $\pi_1$ | $\tau$ | *Base* | *New* | H |
|---|---|---|---|---|---|
| One-Stage Optimization | $\frac{\tau_0}{\tau_1+\tau_0}$ | $\frac{\tau_1 \cdot \tau_0}{\tau_1+\tau_0}$ | 79.3 | **75.1** | **77.0** |
| Two-Stage Optimization | $\frac{\exp(\alpha_1)}{\exp(\alpha_0)+\exp(\alpha_1)}$ | 0.01 | **79.4** | 75.0 | **77.0** |

Table 6: 11 datasets used for base-to-new generalization and cross-dataset transfer

| Dataset | CLASSES | TASK | DESCRIPTION | EXAMPLE CLASSES |
|---|---|---|---|---|
| IMAGENET | 1,000 | Object recognition | Large-scale dataset for object classification with diverse categories | tench, goldfish, great white shark, a tiger shark, etc. |
| CALTECH101 | 100 | Object recognition | Variety of object categories with random background images. | Accordion, Airplane, Brain, Butterfly, Crab, Motorbike, etc. |
| OXFORDPETS | 37 | Fine-grained object recognition | Classification of pet breeds including cats and dogs. | Bengal, Persian, Beagle, American Bulldog, etc. |
| STANFORDCARS | 196 | Fine-grained object recognition | Images of various vehicle types, annotated by model. | 2000 AM General Hummer SUV, 2007 BMW X5 SUV, etc. |
| FLOWERS102 | 102 | Fine-grained object recognition | Classification of various flower species. | Daffodil, Pink Primrose, Tiger Lily, Yellow Iris, etc. |
| FOOD101 | 101 | Object recognition | User-uploaded real-world food photos with varied backgrounds and noise. | Apple Pie, Waffles, Sushi, Chocolate Cake, Bibimbap, etc. |
| FGVCAIRCRAFT | 100 | Fine-grained object recognition | Aircraft images annotated hierarchically by variant, family, and manufacturer. | Boeing 717, DH-82, Falcon 2000, etc. |
| SUN397 | 397 | Scene recognition | Covers diverse scenes including indoor, urban, and natural environments. | Abbey, Airport Terminal, Bedroom, Harbor, Bar, etc. |
| DTD | 47 | Texture attribute recognition | Real-world texture images annotated with descriptive attributes. | Striped, Dotted, Cracked, Fibrous, Scaly, Zigzagged, etc. |
| EUROSAT | 10 | Land use and land cover classification | Satellite images from Sentinel-2 focusing on land use and cover types. | Annual Crop Land, Forest, Highway or Road, River, etc. |
| UCF101 | 101 | Action recognition | Video clips of human actions collected from YouTube in dynamic, real-world environments. | Apply Eye Makeup, Basketball Dunk, Playing Piano, etc. |

## D.2. Details of Dataset

Table 6 lists datasets for base-to-new generalization and cross-dataset transfer. Evaluation across 11 datasets highlights generalizability and efficiency beyond specific tasks or domains.

## D.3. Base-to-New Generalization

Experiments were performed utilizing CLIP with a ViT-B/16 (Dosovitskiy, 2020) backbone. Training was conducted over $50$ epochs with a batch size of 32. The prompt $t$ was optimized using the Adam optimizer with a learning rate of $0.002$ and a weight decay of $5 \times 10^{-4}$. CoA-weights were optimized using SGD with the same learning rate, a momentum of 0.9, and a weight decay of $5 \times 10^{-4}$. The weight for $\mathcal{L}_{\text{CoA}}$ was set to $w = 5.0$, the weight for $\mathcal{L}_{\text{Ent}}$ was set to 10.0, and the margin was set to $d = 0.2$. The prompt length $M$ was set to 16.

## D.4. Few-Shot Class-Incremental Learning

Following Ran et al. (2024), experiments were conducted using CLIP with a ViT-L/14 (Dosovitskiy, 2020) backbone. CoCoA-Mix used prompts of length $M = 2$ per session and accumulated them across sessions, requiring fewer parameters than the baseline except in the final two sessions. Each prompt was trained for specialization within its session, and the final prediction used all prompts from previous sessions. Tuned prompt $t_i$ for each session, along with CoA-weights $\pi_i^{\text{in}}$ and $\pi_i^{\text{out}}$, were stored and reused. To ensure scaling stability across sessions, we used the two-stage optimization strategy for CoA-weights. Considering the number of iterations per session, CoA-weights were optimized for 2 epochs in the initial session, and for 100 epochs in all subsequent sessions. The margin $d$ of the loss $\mathcal{L}_{\text{Ent}}$ was set to 0.1. In the initial session, the out-class set was generated using random words, and in subsequent sessions, it consisted of classes from previous sessions. All other settings followed those of base-to-new generalization.

## D.5. Cross-Dataset Transfer

Experiments used CLIP with a ViT-B/16 backbone. The weight for $\mathcal{L}_{\text{CoA}}$ was set to $w = 7.0$, while the loss weight for $\pi^{\text{in}}$ was set to 2.0. Other settings followed those of base-to-new generalization.

Table 7: Performance comparison on 11 datasets in cross-dataset transfer.

| Method | SOURCE IMAGENET | TARGET AVERAGE | TARGET | | | |
| --- | --- | --- | --- | --- | --- | --- |
| | | | CALTECH101 | OXFORDPETS | STANFORDCARS | FLOWERS102 |
| CLIP | 66.73 | 64.89 | 93.27 | 89.18 | 65.56 | 68.05 |
| CoOp | 69.06 | 59.88 (−5.01) | 91.06 (−2.21) | 86.74 (−2.44) | 59.84 (−5.72) | 62.38 (−5.67) |
| PROGRAD | 70.21 | 62.36 (−2.53) | 92.41 (−0.86) | 87.90 (−1.28) | 62.94 (−2.62) | 66.98 (−1.07) |
| KGCoOp | 70.52 | 64.45 (−0.43) | 93.55 (+0.28) | **89.86 (+0.68)** | 65.61 (+0.05) | 68.33 (+0.28) |
| MAPLE | 69.53 | 65.24 (+0.35) | 93.43 (+0.16) | 89.77 (+0.59) | 65.70 (+0.14) | **71.17 (+3.12)** |
| DEPT | 68.03 | 65.06 (+0.17) | **94.07 (+0.80)** | 89.43 (+0.25) | **65.87 (+0.31)** | 69.93 (+1.88) |
| COCOA-MIX | **70.85** | **65.27 (+0.38)** | 93.46 (+0.19) | 89.07 (−0.11) | 65.59 (+0.03) | 68.72 (+0.67) |

| Method | Target | | | | | |
| --- | --- | --- | --- | --- | --- | --- |
| | FOOD101 | FGVCAIRCRAFT | SUN397 | DTD | EUROSAT | UCF101 |
| CLIP | 85.43 | **24.81** | 62.61 | 44.09 | **48.36** | 67.51 |
| CoOp | 83.29 (−2.14) | 16.71 (−8.10) | 59.40 (−3.21) | 38.44 (−5.65) | 39.24 (−9.12) | 61.66 (−5.85) |
| PROGRAD | 84.37 (−1.06) | 17.10 (−7.71) | 62.67 (+0.06) | 39.87 (−4.22) | 45.39 (−2.97) | 63.98 (−3.53) |
| KGCoOp | 85.83 (+0.40) | 21.18 (−3.63) | 64.84 (+2.23) | 44.30 (+0.21) | 44.64 (−3.72) | 66.39 (−1.12) |
| MAPLE | 86.13 (+0.70) | 23.27 (−1.54) | **66.43 (+3.82)** | 44.83 (+0.74) | 43.73 (−4.63) | **67.93 (+0.42)** |
| DEPT | **86.27 (+0.84)** | 22.10 (−2.71) | 65.77 (+3.16) | 45.53 (+1.44) | 44.00 (−4.36) | 67.60 (+0.09) |
| COCOA-MIX | 85.78 (+0.35) | 24.10 (−0.71) | 63.61 (+1.00) | **46.41 (+2.32)** | 48.18 (−0.18) | 67.78 (+0.27) |

## E. Additional Experimental Results

### E.1. Cross-Dataset Transfer

Table 7 presents the accuracy of the method trained on a single source dataset and evaluated on both source and target datasets. Parenthesized values represent the performance difference relative to zero-shot CLIP. CoCoA-Mix achieves the highest accuracy on the source dataset, highlighting the effectiveness of CoA-loss in specialization. It also achieved the highest average accuracy across 10 target datasets, validating the effect of CoA-weights. For FGVCAircraft, DTD, and EuroSAT, where zero-shot CLIP shows significant performance gaps between source and target datasets, CoCoA-Mix exhibits the most stable and effective transferability, maintaining high performance with minimal accuracy degradation compared with previous methods. This suggests that CoCoA-Mix is particularly effective under significant domain shifts. The relatively lower generalization improvement over base-to-new generalization is discussed in Appendix H.

### E.2. Performance Improvement on Confusing Samples of CoA-loss

We define confusing samples as those misclassified by zero-shot CLIP with a probability gap of 0.5 or less between the correct and incorrect classes. Figure 7 presents the performance of confusing samples from the test data across 11 datasets for the *Base* classes. While slight performance drops are observed on datasets such as Food101, SUN397, and DTD, CoA-loss consistently enhances specialization across most datasets, with notable improvements on OxfordPets, Flowers102, and UCF101. This demonstrates the effectiveness of CoA-loss in handling confusing cases while enhancing the performance across diverse datasets.

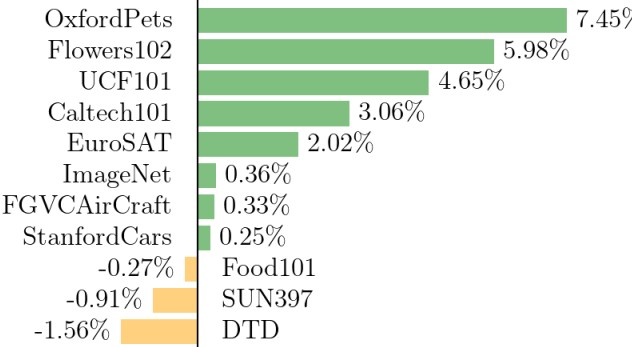

Figure 7: Performance improvement on confusing samples, defined as instances with a probability gap $\leq 0.5$ between correct and incorrect labels or misclassified by zero-shot CLIP.

### E.3. *Base* Performance Comparison According to Loss Function

We compared CoA-loss with four loss functions: focal loss (FL) (Ross & Dollár, 2017), generalized cross-entropy loss (GCE) (Zhang & Sabuncu, 2018), cross-entropy (CE), and mean absolute error (MAE). The results are presented in Figure 8.

FL modifies CE by weighting with $(1 - \hat{p})^\gamma$ as follows:

$$\mathcal{L}_{\text{FL}}(\mathbf{x}, y; \hat{p}) = -(1 - \hat{p}(y))^\gamma \log \hat{p}(y), \tag{16}$$

where $\gamma \geq 0$ is a focusing parameter. FL prioritizes misclassified samples but limits learning from well-classified ones, leading to lower performance in prompt tuning.

GCE generalizes CE and MAE as follows:

$$\mathcal{L}_{\text{GCE}}(\mathbf{x}, y; \hat{p}) = \frac{1 - \hat{p}(y)^q}{q}, \tag{17}$$

where $q \in (0, 1]$ transitions from CE ($q \to 0$) to MAE ($q = 1$). While designed to mitigate the overfitting of CE and slow convergence of MAE, GCE underperforms CE in prompt tuning.

CE, widely used in prompt tuning, is defined as follows:

$$\mathcal{L}_{\text{CE}}(\mathbf{x}, y; \hat{p}) = -\log \hat{p}(y). \tag{18}$$

Its gradients with respect to the similarity $\boldsymbol{s}(y)$ and $\boldsymbol{s}(c \neq y)$ are as follows, respectively:

$$\frac{\partial \mathcal{L}_{\text{CE}}}{\partial \boldsymbol{s}(y)} = -\frac{1}{\tau}\left(1 - \hat{p}(y)\right) \quad \text{and} \quad \frac{\partial \mathcal{L}_{\text{CE}}}{\partial \boldsymbol{s}(c \neq y)} = \frac{1}{\tau}\hat{p}(c), \tag{19}$$

where $\tau$ is the temperature. CE updates gradients based only on individual class probabilities, ignoring inter-class relationships.

Adding MAE to CE, the loss is defined as follows:

$$\mathcal{L}_{\text{CE+MAE}}(\mathbf{x}, y; \hat{p}) = -\log \hat{p}(y) + w\frac{1}{|\mathcal{Y}|}\sum_{l \in \mathcal{Y}} |\mathbb{1}_{l=y} - \hat{p}(l)|, \tag{20}$$

where $w$ controls the contribution of MAE. While CE emphasizes difficult samples, MAE ensures uniform learning over all samples, improving robustness to noise. With optimized $w$, CE+MAE slightly improves performance, but excessive MAE weighting reduces focus on difficult samples, degrading performance.

CoA-loss, similar to MAE, applies only to the correct class $y$ and is combined with CE as follows:

$$\mathcal{L}_{\text{CE+CoA}}(\mathbf{x}, y; \hat{p}) = -\log \hat{p}(y) + w\left(1 - \hat{p}(y)\right), \tag{21}$$

where $w$ controls the contribution of CoA-loss. Unlike CE, CoA-loss considers both $\hat{p}(c \neq y)$ and $\hat{p}(y)$ when updating $\hat{p}(c \neq y)$, focusing on confusing cases where $\hat{p}(y)$ and $\hat{p}(c \neq y)$ approach 0.5. By focusing learning near decision boundaries, CE+CoA achieves the highest specialization performance.

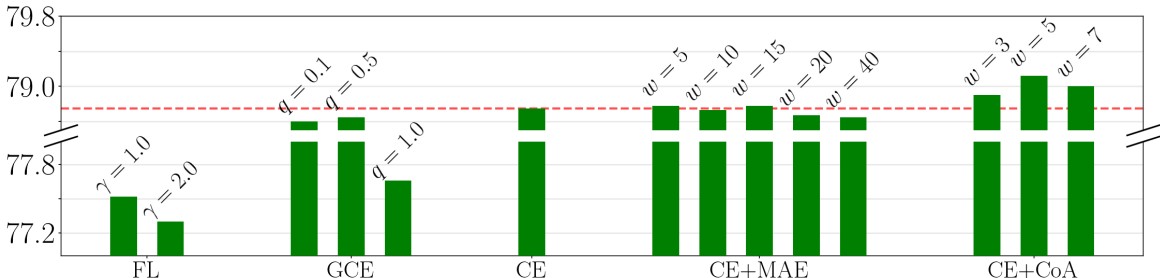

Figure 8: Performance comparison across various loss functions with varying hyperparameters. The hyperparameter setting for each loss is shown above each data.

Table 8: Effect of CoA-weight on *Base* and *New* classes.

| $\pi_i^{\text{in}}$ | $\pi_i^{\text{out}}$ | BASE | NEW | H |
|:---:|:---:|:---:|:---:|:---:|
| ✗ | ✗ | 79.12 | 73.66 | 76.15 |
| ✓ | ✗ | 79.30 | 73.81 | 76.32 |
| ✓ | ✓ | **79.31** | **75.10** | **77.03** |

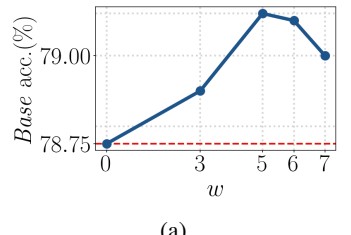
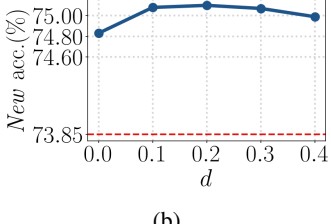

(a)                                          (b)

Figure 9: Sensitivity analysis of hyperparameters. The dotted line indicates baseline performance. (a) Effect of CoA-loss weight $w$ on *Base* accuracy. (b) Effect of margin $d$ on *New* accuracy.

## F. Ablation Studies

### F.1. Effectiveness of Confidence-Aware Weights

To evaluate the effectiveness of CoA-weight, we conducted an ablation study on optimizing weights for *Base* classes ($\pi_i^{\text{in}}$) and *New* classes ($\pi_i^{\text{out}}$) separately and jointly. Table 8 shows that optimizing $\pi_i^{\text{in}}$ for *Base* classes improves performance over the baseline. This indicates that confidence adjustment in the in-class domain provides additional benefits for specialization. When both $\pi_i^{\text{in}}$ and $\pi_i^{\text{out}}$ are optimized, the performance improves further, achieving the best harmonic mean between the *Base* and *New* classes. These results demonstrate that adapting weights for both in-classes and out-classes is crucial for achieving generalization without compromising specialization.

### F.2. Sensitivity of Hyperparameters

We analyzed the sensitivity of our method to hyperparameters by evaluating performance under different configurations. Specifically, we investigated the effect of varying weights $w$ for CoA-loss on *Base* classes and the margin $d$ for CoA-weights on *New* classes.

Figure 9(a) shows the *Base* performance for different $w$ values, with the red dashed line representing the standard cross-entropy baseline. Introducing CoA-loss improves specialization, and tuning $w$ effectively balances the focus on confusing cases, achieving superior performance. Figure 9(b) shows the *New* performance with varying margins $d$. The red dashed line represents the baseline without $\pi^{\text{out}}$. Across all hyperparameter settings, our method outperforms the baseline, indicating that performance is relatively insensitive to the choice of margin $d$.

### F.3. Generation of Unseen Classes

We conducted an ablation study to evaluate the effectiveness of different strategies for generating unseen classes. The study compares three methods: (1) no generation (`None`), (2) generating random strings (`Random String`), (3) generating a total of $N$ classes, with half as random strings and half sampled from the random word API (Rebguns, 2021) (`Random String and Word`), and (4) sampling $N$ words from the random word API (Rebguns, 2021) (`Random Word`). As represented in Table 9, sampling random words yields the best performance, outperforming other methods. This suggests that generating out-classes using semantically meaningful words, rather than arbitrary strings, further enhances generalization. This occurs because semantically meaningful words better align with learned embeddings, providing clearer semantic cues for distinguishing unseen classes, whereas arbitrary strings introduce noise into the embedding space.

Table 9: Ablation study comparing different strategies for generating unseen classes. The table reports accuracy on *New* classes.

| | NONE | RANDOM STRING | RANDOM STRING AND WORD | RANDOM WORD |
|:---|:---:|:---:|:---:|:---:|
| ACCURACY | 74.12 | 75.00 | 75.04 | **75.10** |

# G. Qualitative Visualizations

To better understand model behavior, we provide ScoreCAM (Wang et al., 2020) using the PyTorch library by Gildenblat et al. (2021), comparing CoCoA-Mix with zero-shot CLIP and CoOp. The visualization results are shown in Figure 10. In the *Flowers102* dataset (*New*), CoCoA-Mix more accurately attends to semantically meaningful regions in out-class samples, suggesting that CoA-weights effectively enhance generalization. In the *FGVC Aircraft* dataset (*Base*), CoA-loss focuses more precisely on fine-grained details such as text on airplane wings, outperforming zero-shot CLIP in specialization. These qualitative results demonstrate that CoA-loss and CoA-weights contribute to generalization and specialization, respectively.

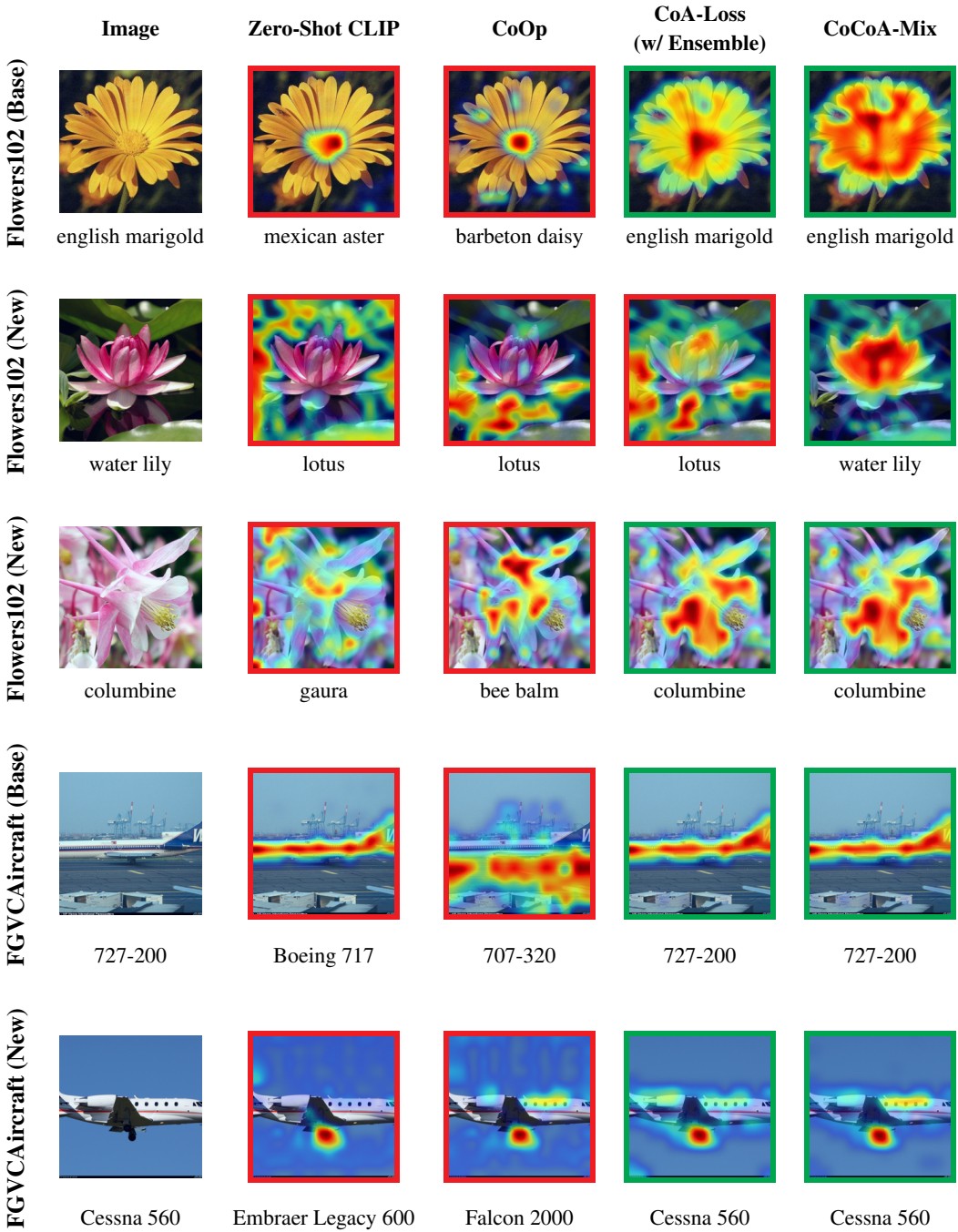

Figure 10: ScoreCAM results on Flowers102 and FGVCAircraft. Red indicates higher activation. The leftmost column shows the image and ground-truth label. Other columns show the ScoreCAM results and predicted labels for each method.

# H. Limitation and Future Work

This study focused on textual prompt tuning. To demonstrate that improved performances in the source domain lead to better performance in the target domain, we adopted the following relation from Nguyen et al. (2022):

$$\epsilon_T(\hat{p}) \leq \epsilon_S(\hat{p}) + \frac{C}{\sqrt{2}}\sqrt{\text{KL}(p_T(\mathbf{z})|p_S(\mathbf{z})) + \delta}, \tag{22}$$

where $\mathbf{z}$ is defined as a visual embedding; $C$ is a constant that bounds $\log \hat{p}(l)$, ensuring each class probability is at least $\exp(-C)$; $p_T$ and $p_S$ are the marginal distribution of $\mathbf{z}$ for the target and source domains, respectively;

This assumes that visual embeddings $\mathbf{z}$ remain consistent across domains, implying similar embeddings for images of the same semantic but differing styles. However, pre-trained CLIP can generate distinct visual embeddings for images with varying styles. As a result, while the proposed method ensures that prompts specialized for the source domain perform well in the target domain from a class perspective, it does not guarantee performance across target domains with different styles. This limitation is reflected in the results, where cross-dataset transfer shows weaker generalization compared with base-to-new generalization. Future work could address this by optimizing visual prompts to align embeddings of images with different styles but the same semantics. Such methods could enhance specialization and generalization across both classes and styles, enabling more robust performance across diverse domains.

