# OpenReview forum: "CoCoA-Mix: Confusion-and-Confidence-Aware Mixture Model for Context Optimization"
_ICML.cc/2025/Conference — ICML 2025 poster_

### Official Review · Reviewer_P3tC · 2025-03-12

**Overall Recommendation:** 2

**Summary:**

The authors explore the prompt tuning problem of vision-language models and propose a training method, CoCoA-Mix, to improve generalization and specialization simultaneously.
CoCoA-Mix consists of a confusion-aware loss to enhance specialization for confusing classes and a confidence-aware temperature strategy that aims to improve generalization.
The authors provide a theoretical derivation for the proposed method.
Moreover, the authors provide experimental results based on CLIP to demonstrate the effectiveness of CoCoA-Mix.

**Claims And Evidence:**

Yes.

**Essential References Not Discussed:**

No.

**Experimental Designs Or Analyses:**

Yes, I checked the results and settings, and the authors provided the source code.

**Methods And Evaluation Criteria:**

Yes, I think the proposed method makes sense for the fine-tuning problem of VLMs.

**Other Comments Or Suggestions:**

please refer to Other Strengths And Weaknesses.

**Other Strengths And Weaknesses:**

Strengths:
1. The authors study prompt tuning problems and propose CoCoA-Mix to enhance generalization and specialization simultaneously.
2. The authors provide theoretical support to make the method more convincing.

Weakness:
1. The writing is not clear. The detailed description of Confidence-Aware Temperature for Generalization without Trade-Offs in the submission is too confusing, which makes it difficult for readers to intuitively understand its content. In addition, the authors did not provide a description of the overall process of the method, and even a description of the overall loss function. (e.g., q only appears once in Eq. 12 but has no specific description.)
2. There are too few works in the related work section, which basically only include a few comparative methods, and cannot give readers a full picture of the field.
3. Lack of baseline CoCoOp. Why is the 4-shot setting used in the experiment in Table 1 instead of the commonly used 16-shot setting?

**Questions For Authors:**

please refer to Other Strengths And Weaknesses.

**Relation To Broader Scientific Literature:**

This submission discusses the weakness of previous prompt tuning methods and proposes CoCoA-Mix to improve generalization and specialization simultaneously.

**Theoretical Claims:**

I check the proof of Theorem 3.2 and I think it is correct.
However, I have doubts about the introduction of the target domain. The paper [1] shows a distribution shift between the target domain and the source domain. This shift generally does not refer to the gap between the empirical error and the expected error.

[1] Kl guided domain adaptation, Nguyen et al., ICLR 2022.

---

> ### Author Rebuttal · Authors · 2025-04-01
>
> ## `Theoretical Claims`
> In our paper, we refer to Eq. 12 in [1], which relates distributional shift to generalization performance:
> $$
> l\_\\text{test} \\leq l\_\\text{train} + \\frac{M}{\\sqrt{2}} \\sqrt{\\text{KL} [p\_T(z)|p\_S(z)] +\\mathbb{E}\_{p\_T(x)}\\left[ \\text{KL}[p\_T(y|x)|p\_S(y|x)] \\right] },
> $$
> where $l\_\\text{train}=\\mathbb{E}\_{(x,y)\\sim p\_S(x,y),z\\sim p(z|x)}[-\\log\\hat{p}(y|z)]$; $l\_\\text{test}=\\mathbb{E}\_{p\_T(x,y)}[-\\log\\hat{p}(y|x)]$; $p\_S$ and $p\_T$ are joint data distributions of source and target domains, respectively; and $z$ is a representations derived from $x$.
> In our setting, $l_\text{train}$ and $l_\text{test}$ correspond to $\epsilon_S(\hat{p})$ and $\epsilon_T(\hat{p})$ by ***Eq. 2***.
> We assume the CLIP encoder yields sufficiently informative representations via large-scale pretraining, satisfying Assumptions 1 and 2 in [1]. This allows us to apply Eq. 12 in [1] to bound the target error $\epsilon_T(\hat{p})$ as follows:
> $$
> \epsilon_T(\hat{p}) \leq \epsilon_S(\hat{p}) + \frac{M}{\sqrt{2}} \sqrt{
> \text{KL}(p_T(\mathbf{z})|p_S(\mathbf{z}))+\lambda
> }.
> $$
>
> We hope this clarifies our method. If we misunderstood your point, we would appreciate further clarification.
>
> [1] Kl guided domain adaptation, Nguyen *et al*., ICLR 2022.
>
> ---
>
> ## `W-1`
>
> **[Confidence-Aware Temperature for Generalization without Trade-Offs]**
>
> We would appreciate clarification on which part of the 'detailed description' was confusing so that we can address it more precisely in the final version.
>
> To clarify, CoA-Temp adjusts the weight between specialized and generalized predictions based on whether the class is in the in-class or out-class domain. The in-class is defined as the set of classes in the training dataset, while the out-class consists of random words not included in the training dataset.
>
> The in-class weight $\alpha_\text{in}$ is optimized using cross-entropy loss on in-class training samples (***Eq. 15***), while the out-class weight $\alpha_\text{out}$ is optimized using an entropy-based loss (***Eq. 16***) under Assumption 3.4, which assumes generalized predictions are more reliable in out-class domains. For details on $\alpha$ optimization and its use during inference, see our responses to Reviewer *AKUY* (`W-2`) and (`W-4`), respectively.
>
> **[Overall process]**
>
> Our method optimizes three components and **each component is optimized simultaneously with a separate loss function and optimizer** :
>
> 1. Prompt $\boldsymbol{v}$ via cross-entropy and CoA loss ***(Eq. 9)***,
> 2. In-class temperature $\tau_{\boldsymbol{v}}^\text{in}$ via a cross-entropy loss ***(Eq. 15)***,
> 3. Out-class temperature $\tau_{\boldsymbol{v}}^\text{out}$ via an entropy-based loss ***(Eq. 16)***.
>
> **[Clarification on $q$ in Eq. (12)]**
>
> $q$ is the probability that a target sample belongs to the in-class domain $\mathcal{D}_T^\text{in}$ (L250–251). It is used only in the theoretical analysis *(Eq. 12)* to decompose the target error into in-class and out-class components.
>
> ---
> ## `W-2`
>
> **[Prompt mixture models in VLMs]**
>
> Recent works such as Allingham *et al*. (ICML'23) [2] and Lu *et al*. (ICML'24) [3] have explored prompt ensembling using hand-crafted prompts or multiple backbones to improve generalization. These methods generally do not target specialization and often require additional inference-time costs. In contrast, our method (1) explicitly improves specialization and generalization and (2) is more efficient than backbone-level ensembling [3], requiring no extra forward passes during inference, and (3) provides a mathematical framework for specialization and generalization in prompt tuning.
>
> We also compared CoA-Temp with the mixing strategy proposed in [3] and the results are provided [here](https://anonymous.4open.science/r/CoCoA-Mix-B466/Table_II-mixing_strategies.png). While the sample-aware weight generator in [3] requires $205,204$ learnable parameters, CoA-Temp achieves superior performance using only two parameters.
>
> [2] J. U. Allingham, *et al*. “A simple zero-shot prompt weighting technique to improve prompt ensembling in text-image models,” in *ICML*, 2023.
> [3] Z. Lu, *et al*. “Beyond sole strength: Customized ensembles for generalized vision-language models,” in *ICML*, 2024.
>
> ---
> ## `W-3`
>
> We have included CoCoOp in our evaluation under the same setting as ***Table 1***. The average performance is as follows:
>
> |  | Base | New | H |
> | --- | --- | --- | --- |
> | CoCoOp | 77.23 | 68.56 | 71.33 |
>
> Our goal is to demonstrate that **CoCoA-Mix performs well even under challenging low-shot conditions**. We found that the benefits of our method are more pronounced when the training data is scarce. Nevertheless, we evaluated the 16-shot setting (results available [here](https://anonymous.4open.science/r/CoCoA-Mix-B466/Table_III-16shots_results.png)). Despite fewer learnable parameters, CoCoA-Mix achieved competitive performance on Base and clearly outperformed others on New.

---

### Official Review · Reviewer_d4Lr · 2025-03-13

**Overall Recommendation:** 3

**Summary:**

This paper proposed a series of techniques to tackle the problem of improving specialization in prompt tuning, including confusion-aware loss, mixture-model using confidence-aware temperature. Extensive experiments are conducted to show the performance of the proposed method.

**Claims And Evidence:**

Yes. The paper claimed that it focuses on the problem of improving specialization for specific domains. Based on this motivation, the authors proposed several techniques to help prompts adapt to the target domains better.

**Essential References Not Discussed:**

None.

**Experimental Designs Or Analyses:**

The experiments are extensive, with a total of 11 datasets tested. I have no doubts about the experimental performance of the method.

**Methods And Evaluation Criteria:**

The method part is basically reasonable, because it focuses on the task-specific ability in prompt tuning and the generalization of universal embedding, and the loss ideas it uses are similar to some uncertainty-aware loss function ideas in the field of domain adaptation. The disadvantage is that the loss function is not novel enough, because similar ideas have already appeared in the traditional field of domain adaptation. The more interesting point is the proof related to the mixture-model and the setting of confidence-aware temperature.

**Other Comments Or Suggestions:**

I don't know if it's possible to do some statistical experiments on the Assumption $3.4$, such as using a certain data set to see if the Assumption $3.4$ is met. Although I intuitively think that the assumption is reasonable, it may need further verification.

**Other Strengths And Weaknesses:**

The experimental performance of the method is very good, which is worthy of recognition. What needs to be discussed is that the performance of $CoCoA-Mix$ is lower than that of $CoA-Loss$ in the three data sets in the experiment, and the author needs to make some clarification and discussion in this regard.

**Questions For Authors:**

None.

**Relation To Broader Scientific Literature:**

This article solves the adaptation problem of prompt-tuning for different tasks to a certain extent, and provides some new analysis for the combination of domain adaptation and prompt-based methods.

**Theoretical Claims:**

I checked the corresponding proof and found no errors. The corresponding proof makes use of Jensen inequality and simple splitting, and its assumption is reasonable.

---

> ### Author Rebuttal · Authors · 2025-04-01
>
> ## `Other Strengths And Weaknesses`
>
> First, the "CoA-Loss" in ***Table 1*** corresponds to a naive ensemble model with a fixed $\\pi=\\{0.5,0.5\\}$. This naive mixture prediction can perform well when the equal weighting is near-optimal for a given dataset. In contrast, CoCoA-Mix optimizes CoA-Temp based on the class domains, thereby enhancing specialization and generalization. In some datasets, the advantage over fixed-weight ensembles may appear limited, particularly under few-shot settings with high variance. To verify this, we evaluated both methods across six random seeds. The results (see below) show that **CoCoA-Mix consistently matches or outperforms the naive ensemble across all datasets when averaged over more trials.** This suggests that the lower performance on the three datasets is mainly due to variance resulting from limited trials.
>
> |  | Caltech101 |  |  | OxfordPets |  |  | EuroSAT |  |  |
> | --- | --- | --- | --- | --- | --- | --- | --- | --- | --- |
> |  | Base | New | H | Base | New | H | Base | New | H |
> | CoA-Loss (w/o mixture model) | 97.2±0.1 | 94.1±0.5 | 95.7 | 94.6±0.7 | 97.3±0.3 | 96.0 | **84.5±0.8** | 62.7±1.5 | 72.0 |
> | CoA-Loss (seed1,2,3) | 97.9±0.1 | **94.5±0.2** | **96.2** | 94.9±0.5 | **97.9±0.1** | **96.4** | 83.4±0.5 | **70.1±2.5** | **76.1** |
> | CoCoA-Mix (seed1,2,3) | **98.0±0.0** | 94.4±0.1 | **96.2** | **95.2±0.4** | 97.6±0.1 | **96.4** | 83.5±0.7 | 69.1±3.1 | 75.5 |
>
>
> |  | Caltech101 |  |  | OxfordPets |  |  | EuroSAT |  |  |
> | --- | --- | --- | --- | --- | --- | --- | --- | --- | --- |
> |  | Base | New | H | Base | New | H | Base | New | H |
> | CoA-Loss (seed1,2,3,4,5,6) | 98.1±0.2 | 93.6±1.4 | 95.8 | 94.6±0.5 | **97.6±0.7** | 96.2 | 84.1±1.4 | **66.9±4.1** | 74.4 |
> | CoCoA-Mix (seed1,2,3,4,5,6) | **98.2±0.2** | **94.3±0.4** | **96.2** | **95.0±0.4** | **97.6±0.3** | **96.3** | **84.4±1.4** | **66.9±3.6** | **74.5** |
>
> ---
> ## `Other Comments Or Suggestions`
>
> We appreciate the suggestion. To empirically verify ***Assumption 3.4***, $\\epsilon\_{T\_v^\\text{in}}(\\hat{p}\_v)\\leq \\epsilon\_{T\_v^\\text{in}}(\\hat{p}\_h)$ and $\\epsilon\_{T\_v^\\text{out}}(\\hat{p}\_h)\\leq\\epsilon\_{T\_v^\\text{out}}(\\hat{p}\_v)$, we conducted a statistical experiment using the CIFAR-100 and the CLIP model. We randomly split the 100 classes into 50 in-class and 50 out-class domains, trained $\hat{p}_v$ using prompt tuning on the in-class subset, and compared it with the zero-shot $\hat{p}_h$ on both domains. This process was repeated over 10 random splits. A box plot summarizing the results is available [here](https://anonymous.4open.science/r/CoCoA-Mix-B466/Assumption3.4-statistical%20experiment-boxplot.png). The results show that $\hat{p}_v$ consistently outperforms $\hat{p}_h$ on in-class samples, while $\hat{p}_h$ achieves better performance on out-class samples.
>
> To assess statistical significance, we conducted one-sided paired $t$-tests on the per-split accuracy gaps. The resulting $p$-values were $p_{\text{in}} = 9.25 \times 10^{-12}$ and $p_{\text{out}} = 2.06 \times 10^{-10}$, both well below the standard threshold of 0.05. These results allow us to reject the null hypothesis and confirm that the inequalities in ***Assumption 3.4*** hold with strong statistical significance. We will include a summary of this verification and the corresponding visualizations in the appendix of the final version.
>
> ---
> ## `Methods And Evaluation Criteria`
>
> Thank you for your feedback. If you could point us to the specific uncertainty-aware loss functions you are referring to, we would be happy to provide a detailed comparison and clarify the differences or contributions of our approach. Additionally, we provide a comparison with focal loss, generalized cross-entropy, and MAE in ***Appendix D.3***, including both formulation and impact on specialization.

---

### Official Review · Reviewer_AKUY · 2025-03-14

**Overall Recommendation:** 2

**Summary:**

This paper addresses the challenge of improving both specialization and generalization in prompt tuning for vision-language models. It proposes a confusion-aware loss (CoA-loss) that refines decision boundaries between confusing classes, enhancing specialization. Additionally, they introduce a confidence-aware temperature (CoA-temp) mechanism within a mixture model to improve generalization by adjusting prediction weights based on confidence levels. The proposed method, CoCoA-Mix, integrates these components and demonstrates superior performance over state-of-the-art approaches in balancing specialization and generalization.

**Claims And Evidence:**

The claim that the provided mathematical framework demonstrates that specialization and generalization can be improved
simultaneously is not supported by convincing theoretical evidence.

The inequality derived in Theorem 3.2 ensures that \( \epsilon_{T}(\hat{p}_t^{\pi}) = \min_i \epsilon_T(\hat{p}_{t_i}) \) when \( \pi \) is optimized. However, it does not guarantee that \( \epsilon_{T}(\hat{p}_t^{\pi}) < \min_i \epsilon_T(\hat{p}_{t_i}) \). Consequently, Theorem 3.2 does not establish that specialization and generalization can be improved simultaneously.

**Essential References Not Discussed:**

No.

**Experimental Designs Or Analyses:**

Yes, there are no outstanding issues in the designs and analyses of this paper.

**Methods And Evaluation Criteria:**

Yes

**Other Comments Or Suggestions:**

Please refer to the weaknesses listed in the previous section.

**Other Strengths And Weaknesses:**

**Strengths:**

1. The proposed mixture model is straightforward and effective in achieving a better trade-off between specialization and generalization in prompt tuning for vision-language models.

2. The empirical results demonstrate the superiority of the proposed methods over state-of-the-art prompt tuning approaches.

**Weaknesses:**

1. The inequality derived in Theorem 3.2 ensures that \( \epsilon_{T}(\hat{p}_t^{\pi}) = \min_i \epsilon_T(\hat{p}_{t_i}) \) when \( \pi \) is optimized. However, it does not guarantee that \( \epsilon_{T}(\hat{p}_t^{\pi}) < \min_i \epsilon_T(\hat{p}_{t_i}) \). Consequently, Theorem 3.2 does not establish that specialization and generalization can be improved simultaneously.

2. The design of confidence-aware temperature requires further clarification: why is the weight $\alpha$ in the mixture model adjusted by optimizing the temperature rather than optimizing $\alpha$ directly?

3. The classification of in-class and out-class domains requires further discussion. In the implementation stage, are the optimization problems in Equations (15) and (16) solved simultaneously or separately in different cases?

4. The details of how to calculate the predictive probability of input images using the tuned mixture model at the inference stage should be discussed in Section 3.

5. Some visualization results in the evaluation section should be included to interpret the superiority of the proposed mixture model over existing methods, such as by visualizing the features on which the proposed algorithm relies.

**Questions For Authors:**

Please refer to the weaknesses listed in the previous section.

**Relation To Broader Scientific Literature:**

The mixture model in prompt tuning is straightforward and has been explored in previous literature; therefore, the contributions of this work to the relevant community are relatively modest.

**Theoretical Claims:**

Yes, the proofs for the theoretical results are correct.

---

> ### Author Rebuttal · Authors · 2025-04-01
>
> ## `W-1`
>
> We agree that Remark 3.3 should state $\\epsilon\_T(\\hat{p}\_t^\\pi)=\\min\_i\\epsilon\_T(\\hat{p}\_{t\_i})$, not $\leq$. However, our main claim regarding the simultaneous improvement of specialization and generalization **relies on Theorem 3.2 and our method, not Remark 3.3**.
>
> The figure [here](https://anonymous.4open.science/r/CoCoA-Mix-B466/Equation-target_error_bounds.png) illustrates how $\alpha_\text{in}$, $\alpha_\text{out}$, and $\\epsilon\_{S\_v}(\\hat{p}\_v)$ determine the upper bound of the target error in the mixture model. CoA-Loss reduces $\\epsilon\_{S\_v}(\\hat{p}\_v)$ and enhances specialization, while CoA-Temp optimizes $\alpha_\text{in}$ and $\alpha_\text{out}$ via ***Eq. 15*** and ***Eq. 16*** to improve generalization. Consequently, CoA-Loss and CoA-Temp jointly reduce the upper bound of $\epsilon_T(\hat{p}_t^\pi)$, thereby enhancing specialization and generalization simultaneously.
>
> **Even without CoA-Temp, $\\epsilon\_{T\_v^\\text{out}}(\\hat{p}\_t^\\pi)=\\min\_i\\epsilon\_{T\_v^\\text{out}}(\\hat{p}\_{t\_i})$ and $\\epsilon\_{T\_v^\\text{in}}(\\hat{p}\_t^\\pi)=\\min\_i\\epsilon\_{T\_v^\\text{in}}(\\hat{p}\_{t\_i})$ guarantee performance at least as good as zero-shot CLIP for out-class domains and learned prompts for in-class domains simply by setting $\alpha_\text{out}=0$ and $\alpha_\text{in}=1$**. This demonstrates that the mixture model can *preserve* both specialization and generalization. With CoA-Loss and CoA-Temp, these can be further *improved* simultaneously. **This is also supported by the empirical results in *Tables 1 and 3***, where CoCoA-Mix consistently outperforms CLIP across both *Base* and *New* domains.
>
> ---
> ## `W-2`
>
> As described in L222–224 (right column), $\tau_{\boldsymbol{h}}$ is fixed to the temperature of the pre-trained CLIP model. Therefore, optimizing $\tau_{\boldsymbol{v}}$ is sufficient to determine both $\alpha$ and $\tau$ via ***Eq. 14***: $\tau=\frac{\tau_h\tau_v}{\tau_h+\tau_v}$ and $\alpha=\frac{\tau_h}{\tau_h+\tau_v}$. The design was chosen to ensure compatibility with CLIP's standard temperature scaling framework.
>
> **We also evaluated the direct optimization of $\alpha$ using a softmax.** The results can be found [here](https://anonymous.4open.science/r/CoCoA-Mix-B466/Table_I-optimization_strategies.png). Specifically, we fixed the pre-softmax logit of $\boldsymbol{h}$ to zero (i.e., $\alpha_h'=0$) and set $\tau$ to the temperature of the pre-trained CLIP model. Table I shows that both approaches yield similar performance, indicating that **our objective is not sensitive to the choice of optimization strategy for $\alpha$.**
>
> ---
> ## `W-3`
>
> The temperatures $\tau_{\boldsymbol{v}}^\text{in}$ and $\tau_{\boldsymbol{v}}^\text{out}$ are optimized simultaneously in the same loop, but each temperature is optimized with its own loss function and optimizer, as described in "[Overall process]" of our response to Reviewer *P3tC*’s comment `W-1`.
>
> ---
> ## `W-4`
>
> At inference time, given the optimized prompt $\boldsymbol{v}$ and the temperatures $\\tau\_{v}^\\text{in}$ and $\\tau\_{v}^\\text{out}$, we compute the mixture weight as follows:
>
> $$
> \\alpha(l)=\\begin{cases}
> \alpha\_\text{in} = \\tau/\\tau\_{v}^\\text{in}&\\text{if $l\\in\\mathcal{Y}\_{S\_v}$} \\\\
> \alpha_\text{out} =\tau/\tau_{v}^\text{out}&\text{otherwise}
> \end{cases}.
> $$
> The predictive probability is then:
>
> $$
> \hat{p}_{t}^{\pi}(l) = \\frac{\\exp\\left((1-\\alpha(l))\\boldsymbol{s}\_{h}(l)/\\tau+\\alpha(l)\\boldsymbol{s}\_{v}(l)/\\tau\\right)}{ \\sum\_{l'\\in\\mathcal{Y}}\\exp\\left((1-\\alpha(l))\\boldsymbol{s}\_{h}(l)/\\tau+\\alpha(l)\\boldsymbol{s}\_{v}(l)/\\tau\\right)}.
> $$
>
> ---
> ## `W-5`
>
> We provide **ScoreCAM visualizations** comparing CoCoA-Mix with CLIP and CoOp to improve interpretability. The visualization results are available [here](https://anonymous.4open.science/r/CoCoA-Mix-B466/Figure_I-scoreCAM.png).
>
> In *Flowers102 (New)*, **CoCoA-Mix attends more accurately to semantic cues in out-class domains**, highlighting that CoA-Temp improves generalization. In *FGVC Aircraft (Base)*, **CoA-Loss attends to fine-grained details** such as wing text more effectively than zero-shot CLIP. These results show that CoA-Temp and CoA-Loss enhance generalization and specialization, respectively.
>
> ---
> ## `Relation To Broader Scientific Literature`
>
> While prompt mixtures have been studied in prior work, our method differs notably in design and objective. Please see our response to Reviewer *P3tC* (`W-2`) for a detailed comparison.
>
> Furthermore, most existing prompt tuning methods are designed for specialization-generalization trade-offs or incremental learning, making it difficult to achieve both within a single framework. In contrast, as shown in ***Table 2***, our method naturally extends to incremental learning without catastrophic forgetting, bridging the two objectives. Therefore, we believe this work makes a meaningful contribution to prompt tuning.

---

### Decision · Program_Chairs · 2025-05-01

**Decision:**

Accept (poster)

**Comment:**

The paper proposes CoCoA-Mix, which aims to enhance specialization and generalization in prompt tuning for vision-language models. It introduces two key components: a confusion-aware loss (CoA-Loss) that sharpens decision boundaries for confusing classes, and a confidence-aware temperature mechanism (CoA-Temp) that adaptively weights predictions in a mixture model based on in-class and out-class confidence. The method is supported by theoretical insights and evaluated across eleven datasets spanning diverse domains.

The paper tackles an important and practically relevant challenge in prompt tuning—balancing task specialization with generalization across domains. Reviewers acknowledged the breadth of the empirical results and the general applicability of the framework. In particular, CoA-Temp was seen as a novel and effective design for improving generalization without compromising specialization, and the empirical performance across base and novel domains was consistently strong. The theoretical formulation was considered mathematically correct, though reviewer AKUY questioned whether the central claim of simultaneous improvement was fully supported by the presented result in Theorem 3.2. Nonetheless, the authors provided clear clarifications and an illustrative breakdown of their bound-based interpretation.

The main concerns come from two reviewers who gave weak rejection. Reviewer AKUY challenged the interpretation of the theoretical result, but the authors clarified that their claim rests on Theorem 3.2 (not the disputed Remark 3.3), and supported it with visual and statistical evidence. Reviewer P3tC pointed out writing clarity issues, missing baselines, and limited related work coverage. The authors responded comprehensively by adding CoCoOp as a baseline, reporting new 16-shot results, and clarifying the optimization process and architectural pipeline. Reviewer d4Lr, who gave a weak accept, found the experiments strong and the theoretical design valid, raising only minor concerns about novelty. Although reviewers did not update their scores after rebuttal, the authors’ detailed and well-substantiated response addresses the core concerns, and the lack of further engagement should not be treated as a sign of unresolved issues.

Considering the above aspects, the AC recommends acceptance to ICML.